# Phytochemical Insights and Industrial Applications of *Camellia japonica* Leaves: A Focus on Sustainable Utilization

**DOI:** 10.3390/nu17213382

**Published:** 2025-10-28

**Authors:** Ezgi Nur Yuksek, Miguel A. Prieto, Antia G. Pereira

**Affiliations:** 1Instituto de Agroecoloxía e Alimentación (IAA), Universidade de Vigo, Nutrition and Food Group (NuFoG), Campus Auga, 32004 Ourense, Spain; ezginur.yuksek@uvigo.es; 2Investigaciones Agroalimentarias Research Group, Galicia Sur Health Research Institute (IIS Galicia Sur), SERGAS-UVIGO, 36213 Vigo, Spain

**Keywords:** camellias, phytochemical analysis, leaf extracts, industrial applications, natural antioxidants

## Abstract

In response to the increasing interest in medicinal plants, researchers have called for the investigation of underexplored, widely distributed species, such as those within the *Camellia* genus. *Camellia japonica* L., though not native to Galicia (NW Spain), is widely cultivated there, primarily for ornamental purposes. Recent phytochemical analyses of *C. japonica* leaves have identified a variety of bioactive compounds, with phenolic compounds being the most abundant, along with carotenoids, terpenoids and fatty acids. These molecules exhibit a range of biological activities, including antioxidant, antimicrobial, anti-inflammatory, and anticancer effects. Nonetheless, certain constituents, such as saponins, triterpenes, and tannins, may exhibit anti-nutritional or mild toxic effects under specific conditions. This review specifically examines the bioactive compounds found in *C. japonica* leaves and their associated health benefits. Furthermore, it underscores the need for innovative approaches to develop sustainable industrial processes for utilizing *C. japonica* leaves, with potential applications in the food, pharmaceutical, and cosmetic industries.

## 1. Introduction

In recent years, the exploitation of naturally occurring bioactive compounds has experienced exponential growth due to their potential in the pharmaceutical, cosmetic, and food industries [1,2]. These compounds have traditionally been extracted from plants, algae, fungi, and microorganisms and have a history of use in various fields, including traditional medicine, nutrition, and, more recently, biotechnology [3,4]. The appeal of these molecules lies not only in their diverse bioactivities, including antioxidant, anti-inflammatory, and antimicrobial properties, but also in their safety and sustainability profile. This is because, unlike chemically synthesized compounds, bioactive molecules used for industrial purposes have not shown problems associated with adverse effects or negative environmental impacts, aspects that have been detected in many synthetic substances [5,6].

As a result, there has been an increasing demand for new food sources and functional compounds, which has driven the exploration of new biological matrices with high nutritional and bioactive potential. This has led, in recent decades, to the research of underutilized species with the aim of diversifying available natural resources and expanding their exploitation in different sectors [7]. In many cases, this underutilization of a raw material with high potential is due to the scarcity of scientific studies supporting its use, even though these matrices have often been traditionally used in food or in home remedies for various pathologies [8]. However, with the growing interest in sustainability and the development of new analytical and extraction technologies, many of these species are being re-evaluated with promising results, having been shown to be a significant source of metabolites with beneficial properties [9,10]. This approach not only allows for the diversification of available natural resources, but also contributes to the conservation of biodiversity and the development or economic improvement of local communities that can benefit from the cultivation and marketing of species widely distributed in their region [11].

In this context, *Camellia japonica* (*CJ*), an evergreen shrub native to Asia, is a species belonging to the *Camellia* genus that is widely distributed in different regions of the world, highlighting the region of Galicia (NW Spain), which constitutes one of the main international benchmarks regarding the cultivation and production of this species, presenting almost 8000 different varieties, a continuous production with stable quality of 1.5 million plants per year (40 million euros), which supposes about 9% of the total Galician agricultural production [12,13]. Despite its widespread distribution, currently, nearly all production of this species is dedicated to ornamental purposes. However, other species of the *Camellia* genus, widely distributed worldwide, such as *Camellia sinensis* and *Camellia oleifera*, have aroused great interest due to their industrial applications, particularly in the production of tea and oil, respectively [14]. Furthermore, extracts of *C. sinensis* have been used in traditional medicine in multiple countries in Asia and East Africa for the treatment of various pathologies, including chronic diseases (i.e., type 2 diabetes), hypertension, hypercholesterolemia, and cataract disease, suggesting significant pharmacological potential [15]. From this perspective, it is important to highlight the evidence suggesting a relatively homogeneous chemical composition within the *Camellia* genus. This opens the door to further investigation into the composition of *CJ*, a species whose chemical characterization remains scarce in the current scientific literature, even though certain parts, such as its seeds and oils, have been traditionally used in China [16,17].

In this regard, the most recent advances in *CJ* research have revealed that this plant represents a matrix rich in a wide variety of bioactive molecules, including significant amounts of various phenolic compounds, carotenoids, triterpenes, flavonoids, and fatty acids, among others [18,19]. The presence of these compounds confers extracts obtained from *CJL* (*C. japonica* leaves) with diverse bioactivities, particularly antioxidant, antimicrobial, anti-inflammatory, and antidiabetic properties [20,21]. This diversity of bioactivities suggests that extracts obtained from *CJL* could have diverse therapeutic applications in the treatment or prevention of various diseases, as well as in the development of new food additives, with special emphasis on functional foods and preservatives. However, edaphoclimatic conditions (such as soil type, climate, and growing conditions), as well as the time of harvest, are crucial factors that can influence the quantity and quality of the compounds present, and therefore the bioactivity of the extracts, which makes homogenization processes necessary to try to avoid these differences as much as possible [22,23]. This knowledge highlights the importance of optimizing harvesting and extraction processes, as well as studying the interactions between the plant’s compounds to maximize its therapeutic potential. Furthermore, studying phytochemical variability under different soil and climate conditions could help develop more efficient and sustainable cultivation strategies, aiming to obtain *CJ* products with consistent, high-quality pharmacological or nutritional properties.

In this review, recent developments in *CJL* chemical characterization are assessed, attempting to associate the presence of these compounds with various bioactivities reported in the extracts. Due to the scarcity of recent studies, it was necessary to resort to relatively old material, with articles dating back up to 3 decades. The information search was carried out through a systematic review using the search engines PubMed, Web of Science, Wiley, and Google Scholar. Keywords used to filter relevant studies include *Camellia japonica*, leaves, bioactives, molecules, composition, and properties. Information regarding official data was directly obtained from original sources, as was the case with the governmental sources Xunta de Galicia and Deputación de Pontevedra (both ascribed to the Spanish government). The inclusion and exclusion criteria were selected according to the Preferred Reported Items for Systematic Reviews and Meta-Analyses (PRISMA) guidelines. A thematic synthesis was then conducted, allowing for the identification of common trends, discrepancies, and gaps in the literature, offering a clear overview of the current state of knowledge, and identifying key areas for future research.

## 2. Search Strategy and Data Collection

The initial search strategy aimed to include only recent publications to provide an updated overview of the phytochemical profile and bioactive potential of *Camellia japonica* leaves. Nevertheless, the limited availability of comprehensive data required broadening the search period to encompass studies published as early as 1980. Relevant studies were retrieved from Scopus, Web of Science, and PubMed using combinations of the terms “*Camellia japonica*,” “leaves,” “phytochemicals,” “bioactive compounds,” “industrial applications,” and “biological activity.” Only peer-reviewed articles in English presenting experimental or analytical information on *CJ* leaves were retained. Publications focused exclusively on other plant parts, non-English sources, and non-peer-reviewed materials were excluded. Following PRISMA recommendations, duplicates were removed and the remaining records were screened through titles, abstracts, and full texts to ensure the inclusion of studies providing reliable and relevant evidence.

## 3. Phytochemicals Isolated in *Camellia japonica* Leaves

A thorough phytochemical analysis allows for the identification of a broad spectrum of bioactive compounds in *CJL*, which could have applications in both the food and pharmaceutical industries, including their use as additives [19]. However, phytochemical composition varies notably according to environmental and seasonal conditions. For example, different studies have reported that variations in temperature, light intensity, soil type, and nutrient availability affect the biosynthesis of key metabolites such as polyphenols, flavonoids, and saponins [24,25]. In addition, leaf developmental stage plays an important role, as young, expanding, and fully mature leaves often display distinct metabolite profiles. Studies comparing leaf samples from different geographical origins of other *Camellia* species have also reported marked variations in antioxidant and antimicrobial activities, highlighting the strong link between environmental, seasonal, and developmental factors and bioactive potential [26,27]. These observations underscore the need for standardized sampling protocols and systematic monitoring of plant development to identify optimal harvest times for specific metabolites, which will ultimately support the production of extracts with greater consistency and reproducibility.

This section will focus on reviewing the phytochemical profile of *CJL*, highlighting the most relevant compounds and their potential applications in these sectors. In this context, the major compounds identified in the study matrix are grouped into three main groups: phenolic compounds, terpenoids, and pigments, which are characterized by various associated bioactive properties. In addition, a series of minor compounds have been identified, including vitamins and fatty acids, especially unsaturated fatty acids. Regarding nutritional composition, there is no scientific evidence available in the current literature, although it is expected that this matrix does not contain a large protein content.

### 3.1. Phenolic Compounds

Phenolic compounds encompass a large group of molecules present in a wide variety of plants and algae, in which they are essential compounds for their growth and survival [28]. This type of compounds are characterized by having a basic structure formed by at least one aromatic phenolic ring with one or more hydroxyl substituents that can be highly polymerized [29,30]. This level of polymerization, along with its level of substitution, allows phenolic compounds to be classified into different classes and subclasses. In the case of *CJL*, the identified phenolic compounds belong to the groups of phenolic acids, phenylpropanoids, flavonoids, phenolic glycosides, tannins, and other minor compounds such as dihydrochalcone glycosides and phenolic glycosides (Figure 1). Phenolic acids are the simplest phenolic compounds in nature. Examples of compounds of this class identified in *CJL* include gallic acid and 4-hydroxyphenol derivatives. The gallic acid content can be increased by fermenting the leaves, a process used to produce some types of *C. sinensis* tea, which results in a lack of bitterness and dark coloration in these teas [31].

Another class of phenolic compounds present in this matrix are phenylpropanoids. Phenylpropanoids are a class of phenolic compounds derived from the amino acid phenylalanine and characterized by a basic C6-C3 structure consisting of a benzene ring with a three-carbon chain. This class of compounds includes subclasses such as hydroxycinnamic acids, lignans, flavonoids, and stilbenes [32]. Among lignans, eugenol stands out as one of the most abundant compounds in *CJL* [33]. The main flavonoids are quercetin, rutin, epicatechin, and their derivates [19,34,35,36]. Within this subclass of phenolic compounds, a group of molecules exclusive to the *Camellia* genus, called camellianosides, has also been identified in significant concentrations. Camellianosides are flavonol glycosides, specifically quercetin-3-*O*-*β*-d-xylopyranosyl-(1→3)-*O*-*α*-l-rhamnopyranosyl-(1→6)-*O*-*β*-d-glucopyranoside. The interest in these molecules lies in their high antioxidant capacity, which is superior to that of reference antioxidants such as L-cysteine and L-ascorbic acid [35].

Phenolic glycosides (Figure 1) are a class of phenolic compounds composed of a phenolic nucleus linked to one or more sugars via glycosidic bonds. Prominent examples include 1-*O*-*β*-D-(6-*O*-*p*-hydroxybenzoyl) glucopyranoside, heterophylliin A, and casuariin [34,37]. Tannins have also been identified in *CJL* (Table 1), specifically camelliatannins. These molecules have as their basic structure an epicatechin nucleus, to which a C-glucosyl ellagitannin unit is attached [38]. The main camelliatannins described include compounds A, B, C, D, F, G and H [38,39,40]. In addition, it was also possible to identify proanthocyanidins and procyanidins in *CJL* [36]. Other minor phenolic compounds in this matrix include phloretin 2′-*O*-*β*-D-glucopyranoside and camellianoside [34,35].

In conclusion, the leaves of *CJ* are rich in phenolic compounds, which contribute to their notable antioxidant properties. The total concentration of phenolic compounds in the leaves has been reported to vary depending on factors such as environmental conditions and plant maturity. The phenolic compound content is higher in young leaves than in mature leaves, reaching a value of up to 74.30 mg/g dry weight, with a lower content in other parts of the plant [41]. These findings underscore the potential of *CJ* leaves as a valuable natural source of bioactive phenolic compounds.

### 3.2. Pigments

The pigments present in *CJL* are divided primarily into two large groups: chlorophylls and carotenoids. Each of these groups has a unique chemical composition that gives it specific properties for both the plant and the development of various applications in medicine and industry [45]. However, it is important to note that, although not all compounds contributing to plant coloration are classified as pigments, certain phenolic compounds, such as flavonoids, can exert a minor influence on *CJL* coloration [13].

#### 3.2.1. Chlorophylls

In *CJL*, the pigments found in the highest concentrations are chlorophylls, which are responsible for the characteristic green coloration of these leaves. This group plays a fundamental role in the process of photosynthesis, being responsible for the absorption of light in the red and blue wavelengths. Therefore, they are essential molecules for the transformation of light energy into chemical energy for the plant [46]. The types of chlorophyll present in *CJL* include chlorophyll a and chlorophyll b [47]. The results indicate that the distribution of chlorophyll in the chloroplasts of different *CJL* tissues varies according to adaptation to different light intensities. While spongy tissue is adapted to capture light under low-intensity conditions, palisade tissue is better prepared to handle high light intensities, which may also influence the relative amount of chlorophyll in each chloroplast type. This suggests that spongy tissue has a higher amount of chlorophyll a/b, which could be related to a greater capacity to capture light under low light intensity conditions [48]. Based on the greater absorption in spongy tissue and the efficiency in capturing direct light in palisade tissue, *CJ* cultivation conditions could be modified to optimize chlorophyll production, adjusting factors such as light intensity and light distribution in the different microenvironments of the plant, thus favoring greater photosynthetic efficiency [49]. Therefore, the concentration of chlorophyll a and b will depend on environmental conditions, such as light intensity and climate.

#### 3.2.2. Carotenoids

Carotenoids are lipophilic compounds derived from isoprene, characterized by a linear or cyclic chain structure of isoprene units with conjugated double bonds. This configuration gives them optical properties that determine their color, which varies from yellow to red, depending on the chain length and the presence of functional groups. Furthermore, the absence or presence of oxygen in their structure allows carotenoids to be classified into carotenes and xanthophylls. Carotenes are hydrocarbon carotenoids without oxygen atoms. The main carotenoids identified in *CJL* are *β*-carotene and *α*-carotene (Table 2). Xanthophylls, on the other hand, are oxygenated carotenoids that include compounds such as lutein, zeaxanthin, neoxanthin, and violaxanthin [3,13,21]. The major xanthophylls in *CJL* are lutein, flavoxanthin, luteoxanthin, neoxanthin, fucoxanthol, violaxanthin, and pheophorbide b and a [47].

These structural differences determine the color of the compound and, therefore, of the raw material. Furthermore, these structural differences mean that the biological functions and distribution in plant tissues of the various carotenoids reported in *CJL* are significantly different. For example, xanthophylls are involved in protection against damage from excessive light, while carotenes are precursors of vitamin A and have antioxidant properties [55]. These properties determine the distribution of these compounds within the leaf, concentrating them in regions where their functional role is most significant, mainly in the cell membrane and cell wall, and exhibiting hydrophilic characteristics [47].

The concentration of carotenoids in *CJL* increases with the maturation of the leaves due to the biosynthesis of these pigments from isoprenoid precursors, such as geranylgeranyl pyrophosphate (GGPP). This allows carotenoids such as *β*-carotene to be formed through various enzymatic reactions, such as the action of phytoene synthase, which contributes to greater protection against oxidative stress and improved photosynthesis under intense light conditions [56,57]. This developmental pattern suggests that determining the optimal harvest time of *CJL* is crucial for maximizing carotenoid yield. Evaluating leaves at the onset of senescence, when they display more intense orange coloration, may be of interest. Leaf senescence involves a series of complex metabolic processes that affect the carotenoid composition of leaves. During this process, chlorophyll degradation can release previously masked carotenoids, such as *β*-carotene, lutein, and zeaxanthin, compounds of great importance in the food industry due to their antioxidant properties and the benefits associated with their consumption, such as eye protection [58,59]. However, it is crucial to distinguish between functional intact carotenoids and chlorophyll degradation products, such as fluorescent red chlorophyll catabolites, which lack significant antioxidant activity [60,61,62]. Accurate identification of these compounds is essential for assessing the nutritional value of senescent leaves. Furthermore, commercial practices such as the fermentation of leaves from the same species (e.g., *C. sinensis*) have been shown to improve the sensory and nutritional properties of final products [63]. Although the effect of fermentation on the carotenoid content of *CJL* leaves has not been widely studied, it is possible that this process alters the carotenoid composition, either by degrading them or, in some cases, by releasing bound forms that could increase the extractable fraction. For example, it has been established that during tea processing in *C. sinensis*, carotenoids undergo complex transformations, including autoxidation, photooxidation, thermal degradation, and enzymatic cleavage catalyzed by carotenoid cleavage dioxygenases. These processes play a key role in determining both the flavor, through the formation of volatile carotenoid derivatives, and the quality, as reflected in the color of the final product [63]. Therefore, exploring the controlled fermentation of *CJL* leaves could offer dual benefits: improving organoleptic properties and possibly affecting carotenoid availability, warranting further experimental investigation.

### 3.3. Terpenoids

Terpenoids constitute another significant class of secondary metabolites present in *CJL* (Table 2). These compounds are characterized by their involvement in the plant’s defense and adaptation mechanisms and, therefore, have a high bioactive potential. This group of molecules is made up of a broad class of organic compounds derived from isoprene units, whose repetition gives rise to the different subclasses of terpenoids [64]. In the case of *CJL*, the subclasses identified were monoterpenoids, triterpenoids, sesquiterpenoids, and saponins. The most abundant compounds were the triterpenoids, particularly squalene and lupeol, which stand out due to their high concentrations and demonstrated anticancer and anti-inflammatory potential [50]. Significant concentrations of saponins, comprising over 10% of the dry weight of *Camellia* material, are also noteworthy for their bioactive potential. More than 38 distinct saponins have been identified in this species [20,65]. Among these saponins, the camellidin family is particularly notable for its high antioxidant, anti-inflammatory, and antifungal potential [51]. This type of molecules is exclusive to the *Camellia* genus. They are conformed with 3*β*-hydroxy-18*β*-acetoxy-28-norolean-12-en-16-one (camellidin I) or 3*β*, 8*β*-dihydroxy-28-norolean-12-en-16-one (camellidin II) as possible aglycon moieties, and d-glucuronic acid, d-glucose and two moles of d-galactose as the sugar moieties [66]. Other saponins found in *CJL* include camellisoides (A, B, E, and G), molecules that are also exclusive to the *Camellia* genus [20]. Monoterpenoids, such as eucalyptol and 3-cyclohexene-1-methanol, were also identified, reaching concentrations of up to 1.18%. Additionally, sesquiterpenoids, including patchouli alcohol, santalol, epicurzerenone, caryophyllene, and isoledene, were detected [50]. Patchouli alcohol was determined as the majority sesquiterpene in *CJL* (3.49%) and is characterized by its anti-inflammatory potential [50].

In conclusion, the terpene profile of *CJL* reveals a diverse array of monoterpenoids, triterpenoid, and sesquiterpenoids, which contribute significantly to the plant’s bioactive properties. Notably, compounds such as lupeol, squalene, and patchouli alcohol are present in considerable concentrations, suggesting their potential roles in the plant’s therapeutic effects. The presence of other terpenoids, including santalol, epicurzerenone, and caryophyllene, further highlights the complex chemical composition of *CJL*, emphasizing their potential applications in pharmacological and medicinal fields.

### 3.4. Minor Compounds

In addition to polyphenols, terpenoids, and pigments, *CJL* are recognized as a natural source of various other compounds (Table 2). These substances are typically present in much lower concentrations within the species, as exemplified by vitamins, minerals, biosugars, and other biological active compounds. The main biosugars present in *CJL* are hexoses and pentoses such as fructose, glucose, and sucrose [53]. These sugars are key components in cellular structures, such as cell walls, where they are associated with cellulose, hemicellulose, and pectin [67]. Furthermore, the concentration of these sugars has been shown to vary significantly between cultivars, as well as with growing conditions, suggesting that these compounds have an adaptive function in the face of different types of stress [68,69]. *CJL* is also a source of fatty acids, mainly saturated fatty acids such as palmitic acid, tridecanoic acid, myristic acid, pentadecanoic acid, heptadecanoic acid, and stearic acid [18]. This makes *CJL* a matrix with low applicability for obtaining fatty acids, especially when compared to other parts of the plant, such as seeds or flowers, which have been shown to have a fatty acid profile with health benefits [70].

Other minor compounds present in *CJL* include vitamin E, which has been shown to exhibit anticancer and anti-inflammatory effects [50,52]. *CJL* are also rich in minerals, with significant contents of phosphorus, calcium, potassium, sodium, iron, manganese, zinc, aluminum, and copper [53,54]. The concentration of these compounds is higher than that reported in other conventional vegetables (i.e., rye, barley, beans) [71]. Moreover, its concentration can be increased by delaying the harvesting or by supplementing the plant with an aluminum solution [53,54]. Regarding amino acids content, the most abundant amino acids in *CJL* are aspartic acid, glutamic acid, histidine, and alanine, whose concentration declines as the harvesting time is delayed [53].

Despite the relatively low concentrations of these compound groups in *CJL*, they play a crucial role in the plant’s overall biochemical diversity and have attracted significant interest due to their potential antioxidant, antimicrobial, and anti-inflammatory properties. Although their levels are lower than those of primary metabolites, their bioactivity indicates a possible contribution to the plant’s defense mechanisms and metabolic flexibility. Further investigations are required to elucidate the interactions and synergistic effects of these minor compounds, as well as to explore their potential applications in pharmaceuticals and nutraceuticals.

### 3.5. Potentially Detrimental or Anti-Nutritional Compounds

Although *CJL* are a rich source of beneficial phytochemicals, it is important to acknowledge some substances present in this raw material may have anti-nutritional effects or safety implications if present in high concentrations or used in concentrated formulations. Among these, tannins, triterpenes and saponins, described in Section 3.1 and Section 3.3, stand out for their ability to interact with other molecules (i.e., proteins, lipids, and minerals) to form complexes, which can reduce the bioavailability and absorption of essential nutrients [72,73]. Previous studies have also reported that high doses (>30 µg/mL) of triterpenes and saponins are linked to gastrointestinal irritation and hemolytic effects [74,75,76]. Moreover, some saponins have limited solubility in water, which can pose challenges in their incorporation and distribution within food matrices. This can affect their effectiveness as emulsifiers or stabilizers in certain food systems [77]. Therefore, although these compounds exhibit several beneficial biological activities, their presence in products intended for human consumption or cosmetic applications requires careful evaluation of their concentration and form of administration.

Furthermore, although specific studies on *CJL* are limited, research on related species, such as *C. sinensis*, has shown the accumulation of heavy metals (i.e., Hg, Pb, Cd), as well as other potentially toxic elements such as arsenic and aluminum, can accumulate in their leaves, particularly under conditions of environmental pollution [78,79,80]. Therefore, these concentrations may vary between different plantations and growing conditions, including the use of fertilizers, making it necessary to evaluate each case individually and compare the measured levels with the safety thresholds established by relevant health authorities. For example, the WHO has set a provisional tolerable weekly intake (PTWI) of mercury of 1.6 µg/kg body weight, considering concentrations exceeding 1 mg/kg in plant tissues hazardous [81]. Permissible levels of lead in plants are set at 2 mg/kg dry weight, while cadmium is recommended to not exceed 0.02 mg/kg dry weight in plant materials. Arsenic, a known carcinogen, occurs naturally in plants at concentrations ranging from 0.02 to 7 mg/kg dry weight, with higher levels indicating potential contamination and associated health risks [82]. This highlights the importance of routine monitoring of residues and metal content in raw materials used in the food, pharmaceutical, and cosmetic industries. On the other hand, although the available toxicological data on *CJL* are limited, some studies have reported antioxidant and cytotoxic effects in cellular models, suggesting that biological activity may depend on the dose, extraction method, and product formulation. Analyses conducted to date indicate that, under controlled conditions, standardized extracts do not present acute toxicity at commonly tested doses [83,84,85]; however, these results cannot be generalized to all preparations or different extraction methods. Therefore, it is recommended that future research incorporate detailed toxicological evaluations, including acute and subchronic studies in in vitro and animal models, quantification of bioactive compounds such as saponins and triterpenes, analysis of potential contaminants and specific complementary assays, such as hemolysis tests or interactions with the cytochrome P450 enzyme system, in order to balance the reported biological benefits with an evidence-based assessment of potential risks and ensure the safety and efficacy of *CJL*-derived products.

## 4. Reported Biological Activities of *Camellia japonica* Leaves

Although research on the chemical composition of *CJL* is still somewhat limited, this plant has been utilized in traditional medicine and cosmetics, particularly in Eastern countries. Its widespread use is largely driven by the biologically active properties of the compounds found within the genus, as well as their favorable organoleptic characteristics. The presence of bioactive compounds such as phenolic compounds, carotenoids, terpenoids, and minor compounds in *CJL* contributes to its potential therapeutic benefits, which have been recognized for centuries. These properties have made the plant a key component in various health and cosmetic applications. Table 3 provides a summary of the key bioactivities reported for *CJL*. Most studies to date have examined *CJL* extracts rather than highly purified individual constituents. This approach limits the ability to attribute bioactivity to specific molecules, as the observed effects may arise from synergistic interactions among multiple compounds. Notably, many investigations include preliminary quantification of total carotenoids and phenolics, reporting significant levels of both classes. These results suggest that the biological activity of *CJL* is more likely driven by the combined contribution of several metabolites than by a single compound in isolation. This information highlights the biological potential of *CJL* beyond its phenolic and flavonoid content, emphasizing its role in free radical scavenging and other bioactive properties.

### 4.1. Antioxidant Activity

*Camellia* species are considered a natural source of antioxidant compounds. This antioxidant capacity is attributed to their significant content of phenolic compounds and carotenoids. Regarding the mechanism of action, most available studies have reported free radical scavenging activity (Table 3). Additionally, *CJL* has demonstrated significant reducing power (IC_50_ 0.95 µg/mL, as measured by the FRAP method) [87]. Other plant parts, such as flowers, have also been shown to exhibit strong inhibitory effects on lipid peroxidation (IC_50_ 22.5 by TBARS method) [21]. Furthermore, several studies on *Camellia* spp. leaves demonstrate that their extracts can enhance the activity of endogenous antioxidant enzymes, such as superoxide dismutase and catalase, which play a key role in protecting against oxidative damage [83,92]. These properties make *CJL* a promising source for the development of preventive therapies against diseases associated with oxidative stress and premature aging.

However, while the mechanisms underlying bioactivity have been extensively studied, the number of studies examining the factors that may modulate this activity is limited, with leaf maturity being one such significant factor. Young *CJL* displayed a stronger ability to scavenge reactive oxygen species, especially hydrogen peroxide and hydroxyl radicals, than their mature counterparts [86]. In another study, the influence of the extraction solvent on the antioxidant capacity of *CJL* extracts was analyzed, observing the best results with the use of methanol and acetone, obtaining DPPH values of 246.56 and 320.17 μg/mL, respectively [88]. These results are consistent with previous studies demonstrating greater antioxidant power in extracts obtained using polar solvents, primarily alcohols. Furthermore, these studies also demonstrated a high antioxidant capacity of fermented *CJL* [31], process used in the production of different types of tea from *C. sinensis* [93]. These results have shown that *CJL* extracts have a high antioxidant power, comparable to other renowned plant extracts [94]. In addition, analyses were performed on extracts that had been purified until a significant degree of purity was reached in the identified compounds. Once separated, the antioxidant capacity of each compound was analyzed, obtaining the best values with rutin, camellianoside, and isoquercitrin (IC_50_ values of 20 µM, 25.8 µM, and 27.9 µM, respectively) [34,35,42]. Future studies should focus on potential synergies and antagonisms between the compounds present in the raw material to develop viable alternatives. Furthermore, it should be noted that the composition of the extracts will depend not only on the extraction technique used, but also on the cultivar and soil and climate conditions. Significant differences in antioxidant activity have been observed between different cultivars [95].

### 4.2. Antimicrobial Activity

*CJL* are also a relevant source of extracts with antimicrobial properties. As with antioxidant activity, an increase in the effectiveness of this bioactivity was observed when polar solvents such as ethanol were used for extraction, allowing for good results in terms of antibacterial, antifungal, and antiviral activity [31]. For example, the methanolic extracts of young *CJL* have been assessed for their antimicrobial properties, demonstrating significant effectiveness against *S. aureus* [96]. Regarding the influence of cultivar, it has been observed that different *CJL* cultivars exhibit varying antimicrobial capacities [83,97,98]. This was demonstrated in a study involving seven varieties (‘Kramer’s Supreme’, ‘C.M. Wilson’, ‘La Pace’, ‘Mrs. Lyman Clarke’, ‘Benikarako’, ‘Fanny Bolis’), where it was found that the ‘Mrs. Lyman Clarke’ cultivar showed the most significant antimicrobial activity against *Enterobacter cloacae* (inhibition zone of 12.5 mm). In contrast, the ‘La Pace’ cultivar exhibited the lowest bioactivity [90]. In another study, *CJL* demonstrates significant potential in inhibiting the microbial growth of various strains, including *Staphylococcus epidermidis*, *Bacillus subtilis*, *Klebsiella pneumonia* and *Escherichia coli*, with inhibition zones ranging from 7.0 to 8.7 mm [31]. In another study, *CJL* extracts were tested against *B. subtilis*, *Streptomyces fradiae*, *Staphylococcus aureus*, *E. coli*, *Pseudomonas aeruginosa*, *Enterobacter* spp. and *Salmonella enterica*, which resulted in inhibition zones ranging from 1 to 15 mm [41]. Based on these previous findings, it can be inferred that *CJL* extracts exhibit greater efficacy against Gram-positive bacteria, with minimal activity observed against Gram-negative bacteria. However, it should be noted that all these studies were conducted with extracts that did not undergo optimization or purification processes. Therefore, future research would be interesting to analyze the compounds that may be responsible for this bioactivity in order to increase the bioactivity of the extracts. Currently, the available studies on this subject are based on the analysis of other parts of the plant, primarily the flowers, in which fumaric acid is suggested to be the compound responsible for the antimicrobial activity [99].

On the other hand, recent studies conducted with *CJL* have applied encapsulation technologies to improve its properties and efficacy. In one of these studies, zinc oxide nanoparticles were developed, which demonstrated a significant increase in the antimicrobial activity of *CJL* extracts, specifically against extended spectrum *β*-lactamase producing bacteria [100]. Another study carried out with gold nanoparticles also showed promising results, with increased antimicrobial activity against *B. subtilis*, *S. aureus*, Streptococcus *faecalis*, *K. pneumoniae*, *P. aeruginosa*, *E. coli*, and the fungus *Candida albicans* [101]. These findings reveal that *CJL* extracts have broad-spectrum antimicrobial potential against Gram-positive and Gram-negative bacteria through both conventional extracts and nanoparticle-based systems. However, a comprehensive investigation of these extracts is required to identify the secondary metabolites responsible for this activity, thereby determining the potential of *CJL* as a source of natural antibiotics.

### 4.3. Anticancer Activity

*CJL* have also been tested against different cancer cell lines, including MCF-7 (human breast adenocarcinoma pleural effusion), A549 (human lung carcinoma cell line), HCT-116 (human colorectal carcinoma cell line), DU145 (human prostate cancer cell line), and NCI-H226 (human lung squamous cell carcinoma line) cells (Table 3). Other cancer cell lines sensitive to *CJL* include Calu-6 (human lung carcinoma) and SNU-601 (human gastric carcinoma) cells, with a strong growth inhibitory concentration below 100 µg/mL [102]. The majority of studies were conducted using unpurified extracts, which consistently exhibited significant antitumor activity at concentrations in the range of 0.85 to 1.25 mM across various lung and colon cancer cell lines [42]. According to chromatographic studies, this intense antitumor activity can be attributed to the presence of triterpenoids and flavonol glucosides, such as camellioside A and B [20]. Another study reported that triterpenoid saponins isolated from *CJL* effectively suppressed cell proliferation in the MCF-7 breast cancer cell line at a concentration of 100 µg/mL [20]. It has also been reported that lupeol, a biologically active compound isolated from *CJL*, exhibits potent anticancer effects [50]. These findings suggest that *CJL* and its natural compounds hold potential as anticancer agents for the treatment of various cancer types. Moreover, the antioxidant and anti-inflammatory properties reported for these extracts may contribute to the development of anticancer activity, as both processes are closely linked to the onset of carcinogenesis [103]. Thus, the development of extracts from *CJL* could offer new therapeutic targets for cancer treatment. However, there is an urgent need for in vivo and interventional studies to adequately evaluate the effectiveness of *CJL* extracts in human patients.

### 4.4. Other Reported Biological Activities

Certain bioactivities of *CJL* remain insufficiently characterized and studied to date. Many of these activities are strongly associated with the antioxidant capacity of the extracts derived from this plant. For example, evidence suggests that *CJL* exhibit anti-inflammatory effects. However, other parts of the plant, such as the oils extracted from the seeds, have demonstrated even greater potential in this regard [104]. In fact, this plant has been traditionally used in Asian medicine for the treatment of inflammatory disorders [18]. Regarding the compounds responsible for this bioactivity, various studies have demonstrated the influence of different terpenoids (e.g., squalene, lupeol, and vitamin E) and saponins, revealing a close relationship between inflammation and tumorigenesis. These properties are attributed to the significant inhibition of nitric oxide production [20,50]. Moreover, the exploration of the biosynthesis pathway of methyl commate B through metabolomics may provide insights into new bioactive compounds with anti-inflammatory potential. Understanding these metabolic pathways could uncover additional components in the plant extracts that contribute to their anti-inflammatory activity by modulating key inflammatory processes [50].

*CJL* have also shown antifungal activity [31]. This bioactivity is attributed to the presence of difference bioactive compounds, with particular emphasis on camellidin I and II, which were effective against *Pestalotiopsis longiseta*, *Pyricularia oryzae* and *Cochliobolus miyabeanus* [51]. Despite this antifungal activity, the presence of the fungus *Rosellinia* sp. PF1022 has been reported in *CJL*. However, the presence of this fungus may be of industrial interest since it produces a metabolite (cyclooctadepsipeptide PF1022A) known for its antifungal activity, which has significant potential for developing parasitic treatments [105].

Another reported bioactivity in *CJL* is their antiviral capability, with activity demonstrated against key viruses such as AIDS [44], porcine epidemic diarrhea virus [20,43], or human immunodeficiency virus type 1 protease [44]. This bioactivity is associated with the presence of various phenolic compounds, particularly highlighting camelliatannin H. However, this bioactivity is lower than the reported for *CJ* fruit [44]. These compounds can interfere with viral replication, modulating immune responses and blocking viral entry into host cells, suggesting their potential as natural therapeutic agents against various viral infections [85,106]. In addition, the occurrence of oleanane-type triterpenes in *CJL* prompts the need to assess if the leaves exhibit antiviral effects against coronavirus as reported for *CJ* flowers [43].

In addition to its antiviral activity, *CJL* has demonstrated skin protection properties against external factors such as UVB rays and reactive oxygen species. Studies show that *CJL* exhibits anti-photoaging effects, reducing the impact of UV-induced premature aging in concentrations ranging from 0.125 to 10.0 mg/mL [86]. Research has also highlighted the ability of *CJL* to reduce the production of matrix metalloproteinase-1, an enzyme responsible for collagen degradation induced by UV exposure, particularly in young leaves [86,107]. Furthermore, oleanane-type triterpene saponins, which have been identified in *CJ* flower buds, are known to suppress melanin production [16]. Given that these saponins may also be present in the leaves, further research is warranted to explore the potential of *CJ* leaves as a source of bioactive compounds with skin-protective properties. These studies offer a scientific perspective on the potential of *CJL* as a biological source of skin-protection compounds, highlighting the need for further research to characterize the specific active constituents.

In addition to its known skin-protective properties, *CJL* have been tested for a range of other bioactivities, although the number of studies on these effects is still relatively limited. Among the identified bioactivities, *CJL* has demonstrated anti-pancreatic properties by inhibiting the pancreatic lipase enzyme, with an IC_50_ value of 0.308 mg/mL, suggesting a role in lipid metabolism regulation [31]. Additionally, *CJL* has shown significant anti-hyperuricemic effects by reducing serum uric acid levels by up to 60%, highlighting its potential in managing hyperuricemia. This effect is attributed to the inhibition of xanthine oxidase activity, a key enzyme involved in uric acid production [19]. The extract has also demonstrated anti-allergic properties, as evidenced by its ability to suppress Syk kinase activation and the production of inflammatory cytokines such as TNF-*α* and IL-4 at a dose of 50 µg/mL [33]. This suggests that *CJL* may offer potential for the management of allergic diseases, autoimmune disorders, and chronic inflammation. Furthermore, *CJL* has been shown to possess antinociceptive effects. In combination with compounds like epicatechin and rutin, the extract reduced neuropathic pain and inhibited the MAPK3 pathway by suppressing microglial activation in the dorsal root ganglion and spinal cord [34,91]. *CJL* also displays neuroprotective effects, with studies indicating an increase in neutral red uptake in neuronal cells, demonstrating its anti-apoptotic properties and mitochondrial protection [87]. These neuroprotective effects were further supported by antioxidant activities in PC12 cells, where phenolic compounds such as quercetin and kaempferol contributed to the observed protection against oxidative stress [87].

As seen for neurodegenerative diseases, other current prevalent diseases such as diabetes have been assessed from a preventive point of view using *CJL.* In one study, consumption of 0.5 mL/day of tea brewed with *CJL* for one month was shown to induce antihyperglycemic and hypolipidemic effects in diabetic rats. Furthermore, this *CJL* intake was associated with a reduction in markers associated with poor diabetes control, such as altered hematobiokinetic parameters (e.g., creatinine, urea, uric acid, aspartate aminotransferase, and alanine aminotransferase) [108]. In another study, a methanolic *CJL* extract was administered to diabetic rats, which resulted in a reduction in blood glucose levels after only one oral administration [109]. This bioactivity is associated with the presence of polyphenols (including theaflavins), catechins, and polysaccharides [110].

In addition, *CJL* have been extensively studied for their potential in preventing other chronic diseases, such as obesity. Studies conducted with Sprague-Dawley rats have demonstrated that *CJL*, when combined with leaves from other Camellia varieties (*C. sinensis*) at low concentrations (1%), significantly reduces both body weight and adipose tissue compared to other tested diets. Additionally, levels of serum and hepatic triglycerides were lowered. This reduction was associated with decreased activity of lipogenic enzymes in the liver. These findings suggest that the mixed tea may have hypotriglyceridemic effects, likely by delaying triglyceride absorption in the small intestine and inhibiting hepatic lipogenic enzymes. The observed bioactivity is attributed to the presence of theasinensins and theaflavins [111].

Overall, these findings suggest that the incorporation of *CJL*-derived products offers a promising approach to address various prevalent health issues. The plant’s bioactive compounds demonstrate significant potential for the development of a range of medical, cosmetic, and nutraceutical products. Therefore, these products could have substantial commercial potential, supported by a growing body of scientific evidence highlighting their beneficial effects. However, it should be noted that most studies on *CJL* bioactivities have been conducted in vitro, with limited evidence from in vivo models. This highlights the need for further research to validate the biological effects under physiological conditions. Additional investigations are necessary to confirm the efficacy and safety of the bioactive compounds, thereby providing a stronger foundation for potential industrial and therapeutic applications. Moreover, the chemical complexity of *CJL* can result in variable extract compositions, which in turn influences consistency, efficacy, and formulation stability. Processing conditions such as temperature, pH, and solvent choice can alter the concentration or activity of key molecules, as well as interactions with other ingredients that might modify sensory or functional properties in the final product. Additionally, variability in raw material quality due to environmental factors or agricultural practices can impact both safety and performance, highlighting the need for rigorous quality control and standardization. Considerations such as solubility, bioavailability, and potential interference with other formulation components must also be addressed to ensure effective incorporation into the different products. Therefore, a comprehensive understanding of these limitations, together with careful optimization of extraction and processing methods, is crucial for the successful industrial application of *CJL*, while ensuring both product quality and safety, and may also guide further research into the specific mechanisms of action of *CJL* extracts, ultimately enhancing their therapeutic potential and ensuring compliance with efficacy and safety standards across diverse consumer markets.

## 5. Applications of *Camellia japonica* Leaf Extracts and Bioactive Compounds

The *Camellia* spp. genus has traditionally been used in various industries, including food, cosmetics, and pharmaceuticals, due to its unique properties and bioactive compounds. Among the most notable species, *C. oleifera* is known for its high-quality oil used in cosmetics, *C. sinensis* is the main source of tea, and *Camellia sasanqua* has been utilized in the perfume industry [14]. However, despite the rich composition and high bioactive potential of *CJ*, its commercialization has historically been limited compared to other species of the same genus. Nevertheless, this species contains a variety of active compounds, such as phenolic compounds, carotenoids, and terpenoids, which could have significant applications in different industrial fields [19,20,31,33,50,86,87,95,101,112,113,114].

In recent years, various scientific studies have promoted the revaluation of *CJ* through the development of new technological applications. Research has identified its potential in the production of dietary supplements, cosmetic ingredients, and pharmaceutical products, as well as in the creation of bioactive materials with antioxidant and anti-inflammatory properties. Thanks to these advances, a range of possibilities has opened up for their inclusion in multiple sectors (Figure 2), which has improved both their commercial value and their integration into sustainable industrial practices.

### 5.1. Food Industry

*CJL* has great potential in the food industry due to its richness in bioactive compounds, such as carotenoids, polyphenols, flavonoids, saponins, and amino acids (Figure 2). Therefore, this matrix is being investigated for the development of various food products, primarily beverages and dietary supplements. Among these is the development of tea-like infusions like the ones produced from the leaves of *C. sinensis*. These infusions have the commercial appeal of being rich in antioxidant compounds and promoting metabolic health, making them a healthy alternative herbal infusion [111,115]. Various *CJL* products are already on the market. For example, in recent years, the Areeiro Phytopathological Station (Pontevedra, Spain) developed a tea from *CJL* marketed under the name “Areeiro Tea”, which opens the door to experimenting with the use of *CJL* in functional beverages.

Furthermore, *CJL* has a rich composition from a nutritional (amino acids, minerals, fiber) and phytochemical (carotenoids, phenolic compounds) perspective, which is why this matrix has been investigated for the development of dietary supplements and food fortification with health-promoting properties, highlighting its antioxidant and anti-inflammatory effects [99,116]. For instance, carotenoids extracted from plant leaves are of particular interest in food and nutrition due to their dual role as natural pigments capable of enhancing or stabilizing the color of food products, and as bioactive compounds with provitamin A activity (in the case of *β*-carotene) and antioxidant capacity that may contribute to oxidative stability and health benefits related to the mitigation of oxidative stress [117,118]. In this context, carotenoid-rich extracts from *CJL* represent a potential source of natural colorants and functional ingredients for incorporation into food formulations, where they could improve both the visual appeal and the nutritional value of products. Nevertheless, systematic evaluations of concentration–response effects, stability during processing (thermal, light and pH conditions), and in vivo bioavailability of *CJL* products remain scarce. In fact, the scientific literature provides limited direct evidence regarding the efficacy of *CJL* extracts in food systems, as most available data rely on comparisons with other leafy matrices rather than on validation within real formulations. Consequently, the technological readiness level (TRL) of *CJL* applications typically falls between TRL 6 and 7, corresponding to pilot-scale validation and pre-commercial evaluation. Although promising results have been reported for stability, sensory properties, and bioactive retention, large-scale industrial adoption remains scarce. Key obstacles include the need for standardized extract composition, optimization of food-grade extraction protocols, and adherence to regulatory requirements for novel ingredients. The current research is progressively addressing these challenges, indicating that several formulations may soon advance to higher TRL stages.

Similar conditions apply to other food formulations developed from different *CJ* organs, such as flowers and seeds, whereas commercial-level TRLs are only achieved in other *Camellia* species (primarily *C. sinensis* for tea production from its leaves and *C. oleifera* for oil extraction from its seeds) [13]. In all these matrices, crude extracts from leaves and flowers have consistently demonstrated remarkable antioxidant activity, likely resulting from the synergistic action of carotenoids, phenolic compounds, and other phytochemicals. These properties suggest a potential role in extending shelf-life and improving the functional profile of foods [31,68]. Other studies have examined additional bioactivities of relevance to food and nutrition, such as the anti-obesity potential of heat-treated *CJL* extracts, which showed strong inhibitory effects on both lipase and *α*-glucosidase activities [119]. These properties indicate the potential for incorporation into functional foods aimed at supporting the prevention of various pathologies. These products include energy bars, smoothies, and antioxidant-enriched yogurts. Furthermore, these antioxidant effects combined with its antimicrobial capabilities make the development of natural preservatives from *CJL* extracts potentially viable, which would reduce the need for synthetic additives [31,116]. In fact, *CJL* has been considered as an improver agent of food products shelf-life, with patents using green tea leaf powder as preservers [120,121].

In recent years, *CJ* extracts have been incorporated into bakery products and pastries due to their distinctive sensory profile and nutritional benefits [122,123]. Furthermore, several restaurants in Spain’s main *CJ* producing region have pioneered the incorporation of this species into a variety of dishes. In this same region, a wide range of products made from *CJ* can be found, including jams, vermouth, liqueur, kombucha, and pastes. Its use in the formulation of these alternatives is due to its properties as a flavor enhancer, providing a slight bitterness and an herbal profile that can enrich the development of healthy products without the need for artificial sugars or synthetic enhancers [122,124].

### 5.2. Cosmetic Industry

The cosmetics industry is one of the main beneficiaries of the advances made with *CJ*. In fact, *CJ* products have achieved relatively high technological readiness levels (TRL 8–9), with several formulations already commercialized due to demonstrated safety, stability, and efficacy in skincare applications. However, most of these currently available *CJ* products use other parts of the plant, such as the seeds, which are used to obtain oils with emollient and skin-regenerating properties. These oils are used in the formulation of anti-wrinkle creams, moisturizing serums, and hair care products [96,125,126].

*CJL* helps strengthen hair, reduce hair loss, and improve the overall health of the scalp. The combination of *CJL* extracts with other hair oils such as castor, black cumin, lavender, rosemary, cedarwood, and lemongrass has been the basis of various patented formulations in recent years, aimed at producing shampoos that protect hair from the effects of the elements, maintain its natural moisture, prevent frizz, and hydrate and restore long-lasting shine [127,128,129]. *CJL* products currently marketed include small-scale productions such as Acemelia’s solid shampoo or internationally marketed shampoos such as Garnier’s Whole Blends, Palmolive, or Oshima Tsubaki shampoo.

In addition, the leaves also exhibit certain potential for the development of other cosmetic applications. Among these, it is worth highlighting the incorporation of their extracts into cosmetic products aimed at combating skin aging and preventing damage caused by exposure to environmental factors such as UV radiation, pollution, and stress [86,130]. This effect is attributed to the high concentration of antioxidants in *CJL*, primarily phenolic compounds, which stimulate cell regeneration and enhance the skin’s repair capacity, thereby improving its texture and firmness [31]. The improved texture achieved with its cosmetic products is also due to *CJL*’s ability to act as a natural moisturizer, helping to maintain skin and hair hydration [125,131]. *CJL* can also be used in the development of cosmetic products with anti-inflammatory properties or that prevent redness or skin reactions in irritated or sensitive skin [127,132]. Other studies proved that *CJL* could be used in the formulation of cosmetic products to control excess oil due to its astringent properties and the fact that it helps regulate sebum production [133,134]. Furthermore, *CJL* can reduce enlarged pores, highly desirable characteristics for the formulation of toners, facial cleansers, and masks [135,136]. Another less developed use includes the formulation of cosmetics with antibacterial properties, making them suitable for products designed to treat skin problems such as acne and other superficial skin infections [96,132].

All these properties make *CJL* a promising ingredient for the formulation of innovative cosmetic products, especially in anti-aging creams, toners, masks, hair care products, and solutions for sensitive or acne-prone skin. For this reason, several prestigious brands have incorporated this matrix into their formulations in recent years. Some examples include Chanel’s lines, formulated with *CJ* extracts, whose active compounds contribute to wrinkle reduction and improved skin elasticity. In fact, Chanel is recognized as one of the pioneers in the incorporation of *CJ* not only in the fashion industry but also in the development of cosmetic products in his line “Chanel Nº1”. Among its most recent advances is the establishment of a sustainable plantation of one of its varieties, The Czar, with the aim of creating a new line of skincare products [137]. Shiseido has also developed various serums and facial oils that incorporate *CJL* as a bioactive component. Meanwhile, companies such as Tatcha and Sulwhasoo use camellia extracts in their luxury cream lines, sold to revitalize the skin and combat the signs of aging. Other cosmetic products from other parts of the plant include hair products or moisturizers made with oil, which have been successfully marketed by companies such as Oshima Tsubaki, Acemelia, Pazo de Rubianes, and Mosteiro de Armenteira. There are also patents in operation (i.e., ES2647357T3) to produce cosmetic and dermatological bars from *CJL* [138].

Most of these products have been developed in the last decade; however, the number of available formulations is expected to continue to increase exponentially. This is due to the compositional similarities of *CJL* with other species of the *Camellia* genus which already have cosmetic products on the market. For example, *C. sinensis* leaf extract in cosmetic products such as sun protection moisturizes [139], moisturizes formulated for the protection of the skin against IR radiations [140], or lip products with improved color-durability and stability [141]. Therefore, *CJL* phytochemical characteristics give competitive advantages for the development of products with specific functional properties, opening new opportunities in sectors such as cosmetics. In this context, *CJL* is positioned as a raw material with high valorization potential.

### 5.3. Pharmaceutical Industry

The potential of *CJL* in the pharmaceutical industry is being explored across various areas, including disease prevention and the development of natural therapeutics. These uses are supported by its traditional use in natural medicine, where it was employed for anti-inflammatory and antimicrobial purposes [13,19]. However, the TRL of *CJL* products in pharmaceutical industry remains moderate (TRL 5–7), as most developments are at the stage of preclinical or early clinical validation, focusing on bioactive compounds with antioxidant, anticancer, and anti-inflammatory potential. Currently, various scientific studies support these applications (Table 3). Many of these applications are due to the presence of compounds present in this matrix have a high antioxidant potential, so these extracts may be useful in preventing oxidative stress, which is linked to the onset of various chronic diseases and cancer [13,31]. In this regard, numerous studies have demonstrated the effectiveness of these extracts against various cancer cell lines, including MCF-7, A549, HCT-116, DU145, and NCI-H226. This activity is attributed to the presence of triterpenoids and other bioactive compounds that could interfere with cell proliferation and promote apoptosis in tumor cells [20].

In addition, *CJL* extracts have been found to have significant anti-inflammatory properties, attributed to their high concentration of terpenoids [50]. These compounds act by inhibiting nitric oxide production and reducing levels of prostaglandin E and tumor necrosis factor-*α*, in addition to decreasing the expression of proinflammatory enzymes such as cyclooxygenase-2 and inducible nitric oxide synthase [142]. These actions suggest certain potential in the development of treatments for inflammatory diseases. These extracts have also demonstrated effectiveness against various pathogenic bacteria such as *S. aureus* and *E. coli*, as well as fungi such as *Candida albicans*. This effect is due to that *CJL* extracts are rich in polyphenols and saponins, which possess the ability to inhibit the growth of these pathogens, suggesting their potential in the development of natural antibiotics and antifungals [31,41,90,143]. These antimicrobial effects open new possibilities for the treatment of infections.

Other less explored potential pharmaceutical applications include allergy medications and drugs with anti-hyperuricemic activity. This is supported by research indicating that ethanolic extracts of *CJL* have inhibitory activity on xanthine oxidase, a key enzyme in uric acid production, and reduce blood uric acid levels in mouse models, suggesting their potential for the treatment of hyperuricemia and gout [19]. Furthermore, the compound okicamelliaside isolated from *CJL* has shown inhibitory effects on mast cell degranulation, a key process in allergic reactions. In vivo studies have shown that both the leaf extract and okicamelliaside can reduce vascular permeability in models of allergic conjunctivitis and decrease sneezing frequency in models of allergic rhinitis, indicating their potential in the development of anti-allergy treatments [144].

In conclusion, despite *CJL* being a promising source of bioactive compounds, there is limited information available on its potential. Therefore, additional research is required to fully explore its industrial applications. The potential of *CJ* in various industries is vast and constantly under investigation. While its use in cosmetics is well established, the food and pharmaceutical industries are still exploring ways to incorporate its benefits into commercially viable products. Growing investment in research and development suggests that in the coming years we could see a significant expansion in the application of this plant in the global market. These may include the development of nutraceuticals, the incorporation of *CJL* compounds into formulations to extend the shelf life of food products, or the formulation of cosmetic products based on its demonstrated bioactive properties.

## 6. Future Perspectives

The development of *CJL*-based applications, alongside the growing demand for natural ingredients and environmentally sustainable products, indicates a strong potential for continued market growth in this sector [145]. This is reflected in the cosmetics market, where natural cosmetics had revenues of around USD 40 billion in 2021 [146]. A similar growth is observed in the food industry, where not only the sustainability of products, but also the development of functional foods, plays a fundamental role, being both of them in high demand [147,148]. The natural food additives market is estimated to experience continued growth of 6.2% annually through 2030, reaching a market value of EUR 9.1 billion in 2021. This represents a significant portion of the global food and beverage market, which is valued at approximately EUR 2.56 billion [149]. This upward trend in the food market opens opportunities for natural additive suppliers in developing countries, including the development of new additives from emerging raw materials such as *CJL*. The market for natural-based pharmaceuticals has also shown significant growth in recent years. Growing consumer preference for more natural and less invasive treatments is driving this trend [150]. With an annual growth rate of 6.8% projected through 2030, the natural medicines market is expected to reach USD 248.6 billion, while the value in 2023 stood at USD 146.6 billion. This market represents a significant portion of the global pharmaceutical sector, which has a total value of approximately EUR 1.3 trillion [151]. This market volume offers new opportunities for natural pharmaceutical producers, especially in developing countries where raw materials for these products are readily available, including exploring emerging ingredients such as *CJL* (Figure 3). However, despite *CJL*’s potential, scientific publications focused on the industrial application of *CJL* extracts remain limited.

The use of *CJL* for different purposes will depend primarily on the extraction method used and the bioactive compound of interest. In this context, optimizing both the extraction process and the purification procedures is crucial, as it ensures the economic efficiency of the process and, therefore, the viability of its exploitation, without compromising the stability or bioactive properties of the target compound(s) [152]. The most commonly used extraction techniques include maceration and distillation [153]. However, the low yields associated with these methods have, in recent decades, driven the development and application of sustainable or “green” technologies, including ultrasound-assisted extraction, enzyme-assisted extraction, microwave-assisted extraction, pulsed electric field-assisted extraction, supercritical fluid extraction, and pressurized liquid extraction [154]. These techniques enabled the extraction of a broad spectrum of biologically active compounds, including various carotenoids (including carotenes and xanthophylls) and phenolic constituents (e.g., (−)-epicatechins, (+)-catechins, gallocatechin gallates, okicamelliasides, camelliosides, camellianosides, chlorogenic acids, rutin, isoquercitrin, and hyperosides) [86,155]. These compounds are widely recognized for their antioxidant and anti-inflammatory properties, which suggest potential applications in food, cosmetics, and pharmaceutical industry [155]. However, the exploitation of these compounds entails certain risks, as many of them are considered heat-labile or photosensitive. Therefore, it is essential to store both the raw material and the extracts under controlled conditions, according to their intended use. The most common treatment consists of drying the leaves by aeration, followed by preserving them under these conditions until processing [156].

Moreover, the later stages of industrialization of *CJ* involve its cultivation, which necessitates the development of new strategies to support large-scale production, primarily through the establishment of plantations. To promote camellia cultivation in emerging regions such as Galicia, it is essential that the site conditions align with the optimal requirements for growth. These typically include shady, humid environments with mild temperatures, fertile, acidic soils, and minimal wind exposure. Beyond geographical factors, seasonality plays a crucial role in plantation success. Sowing should be timed to avoid the growing season (spring) and should preferably occur in soils that are shallow and suitable for planting [157]. Furthermore, it is necessary to investigate how improved cultivation practices and treatments applied to *CJ* (e.g., controlled pesticide use) could enhance production and yield of bioactive compounds of interest. These studies would improve leaf quality and improve crop efficiency and management in a more sustainable manner.

Additionally, since the concentration of secondary metabolites in *CJL* can vary considerably depending on environmental and agronomic factors, systematic crop monitoring is necessary to optimize the production of specific bioactive compounds. Recent studies show that flavonoids, terpenes, and saponins can reach peak levels at different stages of plant development, so determining the most appropriate harvest time for each type of metabolite is crucial for obtaining consistent extracts [158,159]. This integrated approach will enable the production of products with a high degree of standardization and reproducibility.

In summary, a comprehensive approach spanning from laboratory-scale research to open-field cultivation is necessary to facilitate the efficient industrialization of *CJ*, enabling its large-scale utilization as a medicinal plant across various sectors, including cosmetics, food, and pharmaceuticals.

## 7. Conclusions

Despite the ornamental use of *CJ*, *CJL* has been shown to be a source of diverse bioactive compounds, including a wide variety of carotenoids, phenolic compounds, terpenoids, and other minor compounds. Furthermore, scientific evidence demonstrates that extracts obtained from *CJL* exhibit diverse bioactivities, with most available studies focusing on their antioxidant, antimicrobial, and anticancer properties. These multiple bioactivities justify the growing industrial interest in this matrix, with new research directed toward the development of new food products, additives, cosmetics, and pharmaceuticals. In addition, the growing global demand for healthier foods with fewer artificial additives reflects an increasing consumer preference for natural and functional ingredients. This trend presents significant market opportunities for products derived from such sources. As a result, establishing camellia plantations is becoming a key strategy to support the industrialization process, ensuring a sustainable supply of raw materials. Earlier laboratory-scale investigations have given a green light to this initiative and have assured its potentiality also in terms of satisfying production as well as market demands. Future research should prioritize the development of standardized methodologies and targeted investigations to ensure the reproducibility and high quality of extracts, thereby facilitating the industrial application of *CJL*.

## Figures and Tables

**Figure 1 nutrients-17-03382-f001:**
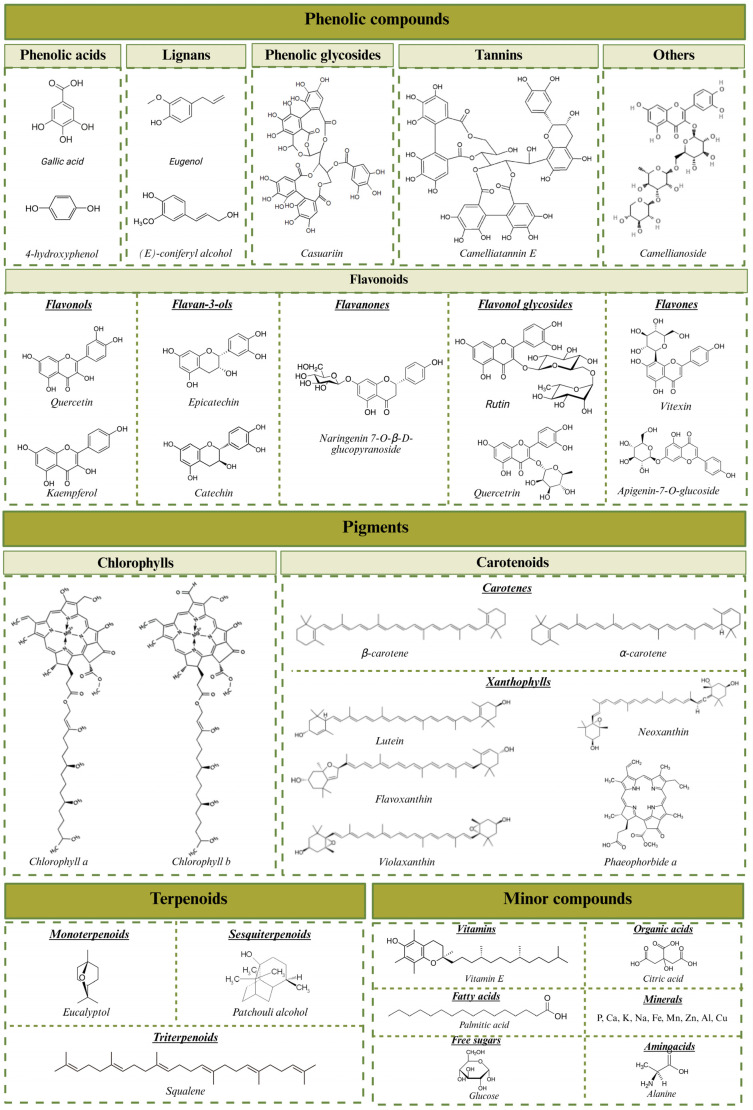
Phytochemicals isolated in *CJL*. Created in BioRender. Prieto, M. (2025), https://BioRender.com/mqbbxy6 (accessed on 26 October 2025).

**Figure 2 nutrients-17-03382-f002:**
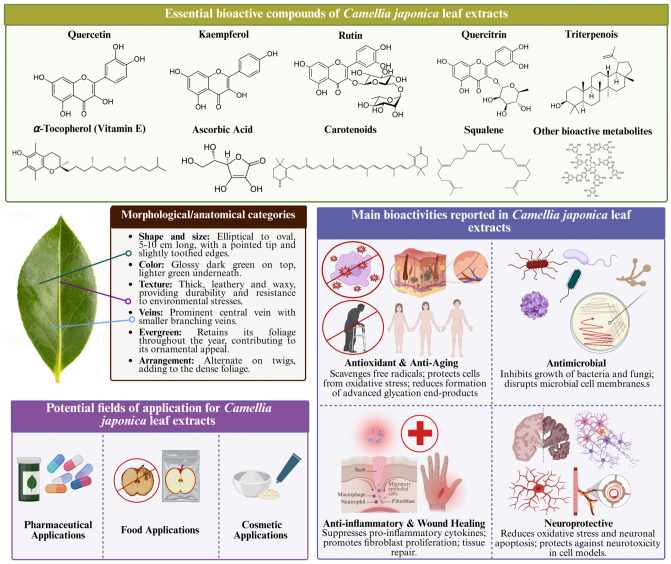
Summary of the major bioactive compounds found in *CJL*, their potential applications and other attributes. Created in BioRender. Prieto, M. (2025), https://BioRender.com/c30a4u2 (accessed on 26 October 2025).

**Figure 3 nutrients-17-03382-f003:**
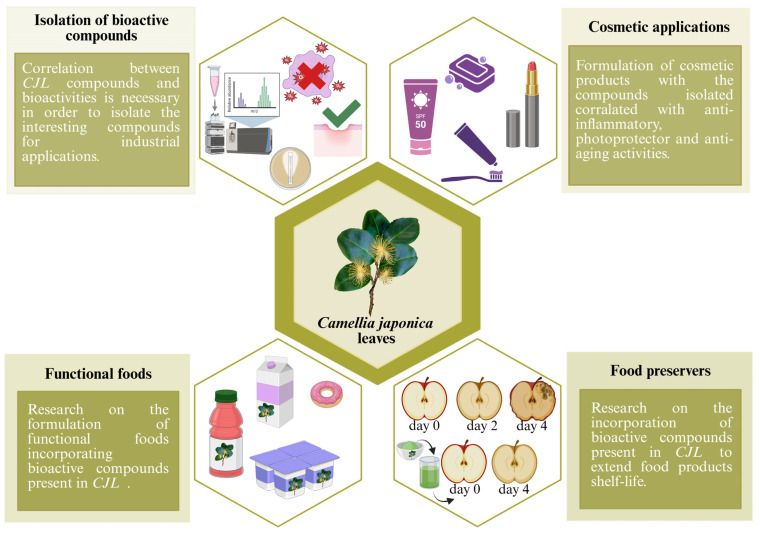
Future perspectives for *CJL* exploitation. Created in BioRender. Prieto, M. (2025) https://BioRender.com/biz37ih (accessed on 26 October 2025).

**Table 1 nutrients-17-03382-t001:** Phenolic compounds reported in *C. japonica* leaves extracts and associated bioactivities.

Subclass	Compounds	Extraction	Identification	Bioactivities	Ref.
**Phenolic acids**					
Benzoic acid derivatives	Gallic acid	SE (MeOH)	GC-MS	Antimicrobial, anti-inflammatory, antioxidant, anticancer	[31,41]
Hydroxyphenols	4-hydroxyphenol derivatives	SE (W, 90 °C)	FABMS, NMR	Antioxidant	[34]
**Phenylpropanoids**					
Lignans	(E)-coniferyl alcohol	SE (W, 90 °C)	FABMS, NMR	Antioxidant	[34]
	Eugenol	UAE (EtOH 100%, 50 °C, 24 h); Y 10%	HPLC	Anti-allergic	[33]
**Flavonoids**					
Flavonols	Quercetin	SE	GC-MS	Antimicrobial, anti-inflammatory, anticancer, antioxidant	[31,42]
	Quercetin; kaempferol	SE (*n*-hexane and EtOH, RT)	GC-MS, NMR	Antioxidant	[18]
Flavan-3-ols	(-)-epicatechin; (+)-catechin	SE (W, 90 °C)	FABMS, NMR	Antioxidant	[34]
		SE (Ace 70%, RT)	HPLC, TLC, NMR	Antioxidant	[36]
	(-)-epicatechin	SE (Ace 70%, 40 °C)	HPLC, NMR, GC-MS	Nd	[37]
		SE	GC-MS	Antimicrobial	[31]
Flavanones	Naringenin 7-*O*-*β*-D-glucopyranoside	SE (W, 90 °C)	FABMS, NMR	Antioxidant	[34]
Flavonol glycosides	Phloridzin; camelliadiphenoside	SE (W, 90 °C)	FABMS, NMR	Antioxidant	[34]
	Rutin; quercetrin	SE (*n*-hexane and EtOH, RT)	GC-MS, NMR	Antioxidant	[18]
	Quercetin-3-*β*-D-glucoside	UAE (EtOH (100%), 50 °C, 24 h); Y 10%	HPLC	Anti-allergic	[33]
	Rutin; hyperoside; isoquercetin	SE (EtOH 60%, RT, 12 h); Y 0.15 mg/g, 0.14 mg/g, 0.09 mg/g, respectively	HPLC	Anti-degranulative, antihistaminic	[35]
	Quercetin 3-*O*-*β*-d-galactopyranoside; quercetin 3-*O*-*β*-d-glucopyranoside	SE (EtOH, RT, 3 days)	NMR, GC-MS	Antioxidant, anti-hyperuricemia	[19]
	Quercetin 3-*O*-*β*-L-rhamnopyranosyl(1→6)-*β*-D-glucopyranoside; kaempferol 3-*O*-*β*-L-rhamnopyranosyl(1→6)-*β*-D-glucopyranoside; quercetin 3-*O*-*β*-D-glucopyranoside; quercetin 3-*O*-*β*-D-galactopyranoside, kaempferol 3-*O*-*β*-D-galactopyranoside; kaempferol 3-*O*-*β*-D-glucopyranoside	SE (W, 90 °C)	FABMS, NMR	Antioxidant, anti-hyperuricemia	[19,34]
Flavones	Quercetin; kaempferol; apigenin	SE (*n*-hexane and EtOH, RT)	GC-MS, NMR	Antioxidant	[18]
**Phenolic glycosides**					
Hydroxyphenol glycoside	4-hydroxyphenol 1-*O*-*β*-D-(6-*O*-*p*-hydroxybenzoyl) glucopyranoside	SE (W, 90 °C)	FABMS, NMR	Antioxidant	[34]
	1,6-di-*O*-*p*-hydroxybenzoyl-*β*-D-glucopyranoside	SE (W, 90 °C)	FABMS, NMR	Antioxidant	[34]
Phenolic glycosides	Heterophylliin A; casuariin	SE (Ace 70%, 40 °C)	HPLC, NMR, GS-MS	Nd	[37]
**Tannins**					
Hydrolyzable tannins	Camelliatannin I and G; pedunculagin; pedunculagin Ox-1	SE (Ace 70%, RT)	HPLC, TLC, NMR	Antioxidant	[36]
	Camelliatannin A, B, and C; tellimagrandin II; strictinin; pedunculagin, 1,2,3,4,6-penta-*O*-galloyl-*β*-D-glucose	SE (Ace 70%, 40 °C)	HPLC, NMR, MS, IR	Antiviral	[37]
	Camelliatannins A, B, C, H	SE	NMR	Antiviral	[13,43]
	Camelliatannins D, F, G		NMR	Antiviral	[38,40]
Condensed tannins	Proanthocyanidins	SE (Ace 70%, RT)	HPLC, TLC, NMR	Antioxidant	[36]
	Procyanidins	SE (Ace 70%, RT)	HPLC, TLC, NMR	Antioxidant	[36]
**Others**					
Dihydrochalcone glycoside	Phloretin 2′-*O*-*β*-D-glucopyranoside	SE (W, 90 °C)	FABMS, NMR	Antioxidant	[34]
Phenolic glycosides	Camellianoside	SE (EtOH 60%, RT, 12 h); Y 0.10 mg/g respectively	HPLC	Anti-degranulative, antihistaminic, antioxidant, antiviral	[35,44]

**Abbreviations:** SE: solvent extraction; UAE: ultrasound-assisted extraction; MeOH: methanol; W: water; EtOH: ethanol; Ace: acetone; RT: room temperature; Y: yield; GC-MS: gas chromatography–mass spectrometry; FABMS: fast atom bombardment mass spectrometry; NMR: nuclear magnetic resonance; HPLC: high-performance liquid chromatography; TLC: thin layer chromatography.

**Table 2 nutrients-17-03382-t002:** Other compounds reported in *C. japonica* leaf extracts and associated bioactivities.

Subclass	Compounds	Extraction	Identification	Bioactivities	Ref.
**Pigments**					
Carotenoids	Lutein; *α* and *β*-carotene; flavoxanthin; luteoxanthin; neoxanthin; fucoxanthol; violaxanthin; pheophorbide b and a	SE (MeOH)	HPLC UV/VIS	Antimicrobial, antioxidant	[47]
Chlorophylls	Chlorophyll a, b	SE (MeOH)	HPLC UV/VIS	Nd	[47]
**Terpenoids**					
Monoterpenoids	Eucalyptol; 3-cyclohexene-1-methanol	SE (MeOH, 25 °C, 8 h), Y 1.18%, 1.79%	GC-MS	Anticancer, anti-inflammatory	[50]
Triterpenoid	Squalene	SE (MeOH, 25 °C, 8 h), Y 27.25%	GC-MS	Anticancer, anti-inflammatory	[50]
	38 different triterpenoids saponins, including: camellioside B, A, E, and G;3*β*,17*β*-dihydroxy-16-oxo-28-norolean-12-en-3-*O*-*β*-d-glucopyranosyl-(1→2)-*O*-*β*-d-galactopyranosyl-(1→3)-*O*-[6-*O*-acetyl-*β*-d-galactopyranosyl-(1→3)]-*β*-d-glucopyranosiduronic acid; 3*β*,16*α*,17*β*-trihydroxy-28-norolean-12-en-3-*O*-*β*-d-glucopyranosyl-(1→2)-*O*-*β*-d-galactopyranosyl-(1→3)-*O*-[*β*-d-galactopyranosy-(1→2)]-*β*-d-glucopyranosiduronic acid	Sox (W, 3 cycles of 3, 2, and 2 h)	UPLC-Q-TOF	Cytotoxic, anti-inflammatory	[20]
Triterpene	Lupeol; methyl commate B	SE (MeOH, 25 °C, 8 h), Y 17.26%	GC-MS	Anticancer, anti-inflammatory	[50]
Sesquiterpenoid	Patchouli alcohol; santalol; epicurzerenone; caryophyllene; isoledene	SE (MeOH, 25 °C, 8 h), Y 3.49%	GC-MS	Anticancer, anti-inflammatory	[50]
Saponins	Camellidin	SE (W, 120 °C, 5 min)	NMR	Antimicrobial	[51]
**Minor compounds**					
Vitamins	Vitamin E	SE (MeOH, 25 °C, 8 h), Y 5.01%	GC-MS	Anticancer, anti-inflammatory	[50,52]
Amino acids	Aspartic acid, glutamic acid, histidine, alanine	-	FS	Nd	[53]
Free sugars	Fructose, glucose and sucrose	-	FS	Nd	[53]
Organic acids	Citric acid, tartaric acid, succinic acid, acetic acid	-	FS	Nd	[53]
Minerals	Phosphorus, calcium, potassium, sodium, iron, manganese, zinc, aluminum, copper	-	FS	Nd	[53,54]
Saturated fatty acids	Palmitic acid; tridecanoic acid; myristic acid; pentadecanoic acid; heptadecanoic acid; stearic acid	SE (Hex, RT)	GC-MS	Antioxidant	[18]
Phthalate ester	Diethyl phthalate	SE (MeOH, 25 °C, 8 h), Y 5.11%	GC-MS	Anticancer, anti-inflammatory	[50]
Ketone	Methyl (3-oxo-2-pentylcyclopentyl)acetate	SE (MeOH, 25 °C, 8 h), Y 27.25%	GC-MS	Anticancer, anti-inflammatory	[50]

**Abbreviations:** SE: solvent extraction; Sox: Soxhlet; MeOH: methanol; W: water; Hex: *n*-hexane; RT: room temperature; Y: yield; GC-MS: gas chromatography–mass spectrometry; UPLC-Q-TOF: ultra-high-performance liquid chromatography–quadrupole time-of-flight mass spectrometry; NMR: nuclear magnetic resonance; HPLC UV/VIS: high-performance liquid chromatography ultraviolet/visible detection; FS: flame spectroscopy; Nd: not determined.

**Table 3 nutrients-17-03382-t003:** Main bioactivities reported in *CJL*.

Phytochemical	Mechanism	Assay//IC_50_	Ref.
**Antioxidant activity**			
Epicatechin	RSA	DPPH//16 mg/mL	[34]
Quercetin 3-*O*-*β-D*-glucopyranoside	RSA	DPPH//16 mg/mL	[34]
Quercetin dihydrate	RSA	DPPH//35.8 µM	[35]
Rutin	RSA	DPPH//20 mg/mL	[34]
Rutin	RSA	DPPH//23.0 µM	[35]
Catechin	RSA	DPPH//20 mg/mL	[34]
Hyperoside	RSA	DPPH//33.1 µM	[35]
Isoquercitrin	RSA	DPPH//27.9 µM	[35]
1,6-di-*O*-*p*-hydroxybenzoyl-*β*-D-glucopyranoside	RSA	DPPH//32 mg/mL	[34]
Phloretin 2′-*O*-*β*-D-glucopyranoside	RSA	DPPH//41 mg/mL	[34]
Camelliadiphenoside	RSA	DPPH//180 mg/mL	[34]
Camellianoside	RSA	DPPH//25.8 µM	[35]
Polyphenols	RSA	DPPH//Nd	[42]
*CJL* extract	RSA	DPPH, RP//38.53 µg/mL, 13.34 µg/µg E	[19]
*CJL* extracts	RSA	DPPH//7.16–18.14 µg/mL	[41]
*CJL* extract	RSA	H_2_O_2_, OH//0.878 and 0.079 mg/mL	[86]
*CJL* extract	RSA	DPPH, FRAP//184.65 µg/mL, 0.95 µg/mL	[87]
*CJL* extract	RSA, neuroprotective	Neuronal cell culture, MTT Assay//125 µg/mL (27.29% inhibition), 250 µg/mL	[87]
*CJL* water and butanol fraction	RSA	DPPH, NSA//92.15% and 95.61%	[42]
*FCJL* ethanolic extract	RSA	DPPH, O_2_, H_2_O_2_, NO//0.22–0.35 mg/mL	[31]
Quercetin, kaempferol, apigenin	RSA	DPPH//Nd	[18]
Rutin, quercetrin	RSA	TLC-DPPH Assay//Nd	[18]
Polyphenols	RSA	UV-Vis Spectrophotometry//Nd	[53]
Lutein, *α*-carotene, *β*-carotene	RSA	HPLC UV/VIS//Nd	[47]
Lutein, *α*-carotene, *β*-carotene	RSA	HPLC UV/VIS//0.21 mg/mL	[31]
*β*-carotene	RSA	DPPH//246.56 mg/mL	[85,88]
Carotenoids	RSA	UV-Vis Spectrophotometry//Nd	[89]
**Antimicrobial**			
FCJL ethanolic extract	AA *Staphylococcus epidermidis*, *Bacillus subtilis*, *Klebsiella pneumonia*, *Escherichia coli*	ADD//7.0–8.7 mm	[31]
*CJL* ethanolic extract	AA *Enterobacter cloacae*	ADD//12.5 mm	[90]
*CJL* extract	AA *B. subtilis*, *Streptomyces fradiae*, *Staphylococcus aureus*, *E. coli*, *Pseudomonas aeruginosa*, *Enterobacter* spp., *Salmonella enterica*	ADD//1–15 mm	[41]
**Anticancer**			
*CJL* butanol and water fraction	Lung and colon cancer cell proliferation inhibition	MTT assay//0.85–1.25 mM	[42]
Triterpenoid saponins	Cell proliferation inhibition	MCF-7//100 µg/mL E	[20]
28-noroleanane glucuronic–galactoside saponin	Cell proliferation inhibition	MCF-7//35.24 μM	[20]
Camellioside A	Cell proliferation inhibition	A549, HCT-116//23.02, 39.22 μM	[20]
Camellioside B	Cell proliferation inhibition	DU145, NCI-H226//32.14, 62.12 μM	[20]
**Other**			
Camellidin I and II	Antifungal. Inhibition of *Pestalotia longiseta*, *Pyricularia oryzae*, *Cochliobolus miyabeanus*	Radial growth method//Nd	[51]
Compound **1**	Antiviral against porcine epidemic diarrhea virus	CPE//0.34, 2.90, and 3.37 µM	[43]
Compound **2**	Anti-inflammatory. NO production inhibition	NO assay//37.99, 31.31, and 28.96% inhibition	[20]
*CJL* extract	Anti-photoaging	UVB, SC, ROS//10, 10, 0.125 mg/mL	[86]
FCJL methanolic extract	Antipancreatic. Hydrolytic reaction of *p*-nitrophenyl butyrate with pancreatic lipase	Pancreatic lipase inhibition activity//0.308 mg/mL	[31]
*CJL* extract	Anti-hyperuricemic. Serum uric acid reduction	XO inhibitory activity, mouse model//60% (2 mg/mL E); 300 mg/kg	[19]
*CJL* extract	Anti-allergic. Inhibition of Syk kinase activation; suppression of TNF-*α* and IL-4	Mouse model//50 µg/mL	[33]
Epicatechin, rutin	Antinociceptive. Reduced CCI-induced punctate allodynia; attenuated dynamic allodynia and spontaneous pain behavior	3 types of allodynia in vivo models//30 mg/kg	[91]
*CJL* extract	Antinociceptive. MAPK activation in the dorsal root ganglion and microglial activation in the spinal cord	3 types of allodynia in vivo models//300 mg/kg	[91]
*CJL* extract	Anti-apoptotic. Mitochondrial protection in neuronal cells	Neutral red uptake assay//106–156% protective effect	[87]

**Abbreviations:** FCJL: fermented *C. japonica* leaf; Compound **1**: 3*β*,16*α*-dihydroxyolean-12-en-28-al 3-*O*-*β*-D-glucuronopyranoside; 3*β*-hydroxy-28-norolean-12,17-dien-16-one 3-*O*-6′-methoxy-*α*-D-glucuronopyranoside; echinocystic acid 3-*O*-[*β*-D-galactopyranosyl(1→2)]-[*β*-D-glucopyranosyl(1→2)-*β*-D-galactopyranosyl(1→3)]-*β*-D -glucuronopyranoside; Compound **2**: camellenodiol 3-*O*-*β*-D-glucopyranosyl-(1→2)-6′′′-*O*-acetyl-*β*-D-galactopyranosyl-(1→3)-[*β*-D-galactopyranosyl-(1→2)]-6′-methoxy-*β*-D-glucuronopyranoside; camellioside B; camellioside A; AA: antimicrobial activity; Nd: not determined; RP: reducing power; O_2_: superoxide radical scavenging assay; H_2_O_2_: hydrogen peroxide scavenging assay; NO: nitric oxide scavenging assay; OH: hydroxyl radical scavenging activity; FRAP: ferric reducing ability of plasma; ADD: agar disk diffusion; RSA: radical scavenging activity; UVB: UVB irradiation on stratum corneum ex vivo; ROS: intra-cellular ROS level; SC: SC carbonyls content; E: extract; XO: xanthine oxidase; NSA: nitrite scavenging assay; CPE: cytopathic effect inhibition assay; MCF-7: human breast adenocarcinoma cell line; A549: human lung carcinoma cell line; HCT-116: human colorectal carcinoma cell line; DU145: human prostate cancer cell line; NCI-H226: human lung squamous cell carcinoma line.

## Data Availability

Not applicable.

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
