# Peer review of "Phytochemical Insights and Industrial Applications of *Camellia japonica* Leaves: A Focus on Sustainable Utilization"

_nutrients, 2025, doi:10.3390/nu17213382_

Round 1
Reviewer 1 Report
Comments and Suggestions for Authors
This review manuscript addresses the growing interest in medicinal plants. The authors describe the characteristics of Camellia japonica L., a plant that is commonly cultivated in Galicia, north-western Spain, despite not being native to the region. They focus on its leaves, which, according to recent research, contain high levels of bioactive compounds, particularly phenols, as well as carotenoids, terpenoids, and fatty acids.
One significant advantage of the manuscript is the way it provides a comprehensive overview of different groups of compounds and their potential biological properties, including antioxidant, antimicrobial, anti-inflammatory and anti-cancer effects. This review synthesises knowledge about the biological activity of these compounds and their practical applications, particularly in the food, pharmaceutical and cosmetics industries.
It is worth noting that the authors emphasise the need to develop innovative, sustainable technologies that make effective use of C. japonica leaves. This is in line with current ecological and circular economy trends.
However, I believe that authors should also critically discuss substances whose effects may be detrimental. For example, does the plant in question contain anti-nutritional substances? Is the use of the plant or leaf extract safe?
Any adverse properties or effects that the plant may have on the body should be mentioned in the summary or in a separate short subchapter.
Additionally, the discussion should cover any issues encountered when using plant leaves in various industries.
Author Response
This review manuscript addresses the growing interest in medicinal plants. The authors describe the characteristics of Camellia japonica L., a plant that is commonly cultivated in Galicia, north-western Spain, despite not being native to the region. They focus on its leaves, which, according to recent research, contain high levels of bioactive compounds, particularly phenols, as well as carotenoids, terpenoids, and fatty acids.
One significant advantage of the manuscript is the way it provides a comprehensive overview of different groups of compounds and their potential biological properties, including antioxidant, antimicrobial, anti-inflammatory and anti-cancer effects. This review synthesizes knowledge about the biological activity of these compounds and their practical applications, particularly in the food, pharmaceutical and cosmetics industries.
Response: We sincerely thank the reviewer for this encouraging feedback. We are glad that the manuscript’s comprehensive overview and synthesis of the biological activities and potential applications of the different compound groups were appreciated. Our main goal was to present an integrated perspective that underscores both the scientific importance and the practical relevance of these compounds in fields such as food, pharmaceuticals, and cosmetics.
The responses to each of your comments are highlighted in red. Additionally, in some responses, you will find text in blue, which indicates how the new text appears in the revised manuscript. Please note that all changes have been made using Track Changes in the manuscript.
It is worth noting that the authors emphasize the need to develop innovative, sustainable technologies that make effective use of C. japonica leaves. This is in line with current ecological and circular economy trends.
Response: We appreciate the reviewer’s positive observation. We are pleased that the emphasis on developing innovative and sustainable technologies for the effective utilization of C. japonica leaves aligns with current ecological and circular economy principles. Highlighting this aspect was indeed one of our key intentions, as we believe it represents a promising direction for future research and practical applications.
However, I believe that authors should also critically discuss substances whose effects may be detrimental. For example, does the plant in question contain anti-nutritional substances? Is the use of the plant or leaf extract safe?
Response: We thank the reviewer for this valuable and constructive suggestion. We fully agree that addressing potentially detrimental substances and safety considerations is essential for a balanced evaluation of C. japonica leaves. Accordingly, we have added a new subsection (Section 2.5: “Potentially detrimental or anti-nutritional compounds”) discussing the presence of compounds that may exhibit anti-nutritional or toxic effects, as well as the current evidence regarding the safety of C. japonica leaf extracts. This addition aims to provide a more comprehensive and critical perspective on both the beneficial and potentially adverse properties of the plant’s phytochemicals.
In the current version, the text reads as follows:
2.5. Potentially detrimental or anti-nutritional compounds
Although CJL are a rich source of beneficial phytochemicals, it is important to acknowledge some substances present in this raw material may have anti-nutritional effects or safety implications if present in high concentrations or used in concentrated formulations. Among these, tannins, triterpenes and saponins, described in Section 2.1 and Section 2.3, stand out for their ability to interact with other molecules (i.e., proteins, lipids, and minerals) to form complexes, which can reduce the bioavailability and absorption of essential nutrients [68,69]. Previous studies have also reported that high doses of triterpenes and saponins are linked to gastrointestinal irritation and hemolytic effects [70,71]. Moreover, some saponins have limited solubility in water, which can pose challenges in their incorporation and distribution within food matrices. This can affect their effectiveness as emulsifiers or stabilizers in certain food systems [72]. Therefore, although these compounds exhibit several beneficial biological activities, their presence in products intended for human consumption or cosmetic applications requires careful evaluation of their concentration and form of administration.
Furthermore, although specific studies on CJL are limited, research on related species, such as C. sinensis, has shown the accumulation of heavy metals (i.e., Hg, Pb, Cd), as well as other potentially toxic elements such as arsenic and aluminum, can accumulate in their leaves, particularly under conditions of environmental pollution [73–75]. This highlights the importance of routine monitoring of residues and metal content in raw materials used in the food, pharmaceutical, and cosmetic industries. On the other hand, although the available toxicological data on CJL are limited, some studies have reported antioxidant and cytotoxic effects in cellular models, suggesting that biological activity may depend on the dose, extraction method, and product formulation. Analyses conducted to date indicate that, under controlled conditions, standardized extracts do not present acute toxicity at commonly tested doses [76–78]; however, these results cannot be generalized to all preparations or different extraction methods. Therefore, it is recommended that future research incorporate detailed toxicological evaluations, including acute and subchronic studies in in vitro and animal models, quantification of bioactive compounds such as saponins and triterpenes, analysis of potential contaminants and specific complementary assays, such as hemolysis tests or interactions with the cytochrome P450 enzyme system, in order to balance the reported biological benefits with an evidence-based assessment of potential risks and ensure the safety and efficacy of CJL -derived products.
Any adverse properties or effects that the plant may have on the body should be mentioned in the summary or in a separate short subchapter.
Response: We sincerely thank the reviewer for this helpful suggestion. To ensure that potential risks associated with Camellia japonica leaves are clearly communicated, we have added a brief statement in the abstract highlighting the possibility of adverse effects. Specifically, the abstract now mentions that certain compounds, such as saponins, triterpenes, and tannins, may have anti-nutritional or mild toxic effects under certain conditions. In addition, as described in our response to the previous comment, a dedicated subsection (Section 2.5: “Potentially detrimental or anti-nutritional compounds”) has been incorporated into the manuscript. This subsection provides a detailed discussion of these compounds, possible environmental contaminants, and current toxicological evidence. These additions provide a balanced perspective, complementing the discussion of beneficial bioactive compounds, and ensure that both the advantages and potential risks of C. japonica leaves are clearly addressed for readers.
In the current version, the text reads as follows:
2.5. Potentially detrimental or anti-nutritional compounds
Although CJL are a rich source of beneficial phytochemicals, it is important to acknowledge some substances present in this raw material may have anti-nutritional effects or safety implications if present in high concentrations or used in concentrated formulations. Among these, tannins, triterpenes and saponins, described in Section 2.1 and Section 2.3, stand out for their ability to interact with other molecules (i.e., proteins, lipids, and minerals) to form complexes, which can reduce the bioavailability and absorption of essential nutrients [68,69]. Previous studies have also reported that high doses of triterpenes and saponins are linked to gastrointestinal irritation and hemolytic effects [70,71]. Moreover, some saponins have limited solubility in water, which can pose challenges in their incorporation and distribution within food matrices. This can affect their effectiveness as emulsifiers or stabilizers in certain food systems [72]. Therefore, although these compounds exhibit several beneficial biological activities, their presence in products intended for human consumption or cosmetic applications requires careful evaluation of their concentration and form of administration.
Furthermore, although specific studies on CJL are limited, research on related species, such as C. sinensis, has shown the accumulation of heavy metals (i.e., Hg, Pb, Cd), as well as other potentially toxic elements such as arsenic and aluminum, can accumulate in their leaves, particularly under conditions of environmental pollution [73–75]. This highlights the importance of routine monitoring of residues and metal content in raw materials used in the food, pharmaceutical, and cosmetic industries. On the other hand, although the available toxicological data on CJL are limited, some studies have reported antioxidant and cytotoxic effects in cellular models, suggesting that biological activity may depend on the dose, extraction method, and product formulation. Analyses conducted to date indicate that, under controlled conditions, standardized extracts do not present acute toxicity at commonly tested doses [76–78]; however, these results cannot be generalized to all preparations or different extraction methods. Therefore, it is recommended that future research incorporate detailed toxicological evaluations, including acute and subchronic studies in in vitro and animal models, quantification of bioactive compounds such as saponins and triterpenes, analysis of potential contaminants and specific complementary assays, such as hemolysis tests or interactions with the cytochrome P450 enzyme system, in order to balance the reported biological benefits with an evidence-based assessment of potential risks and ensure the safety and efficacy of CJL -derived products.
Additionally, the discussion should cover any issues encountered when using plant leaves in various industries.
Response: We thank the reviewer for this suggestion. To address it, we have expanded the discussion in Section 4 to include practical considerations and potential challenges associated with the use of Camellia japonica leaves in various industries. This includes possible limitations related to processing and formulation, such as the stability of bioactive compounds during extraction and storage, potential interactions with other ingredients, and the need to monitor environmental contaminants like heavy metals or pesticide residues. We believe that these additions provide a more comprehensive and realistic perspective on the industrial use of this plant matrix, complementing the discussion of its biological activities and beneficial applications.
In the current version, the text reads as follows:
However, the chemical complexity of CJL can result in variable extract compositions, which in turn influences consistency, efficacy, and formulation stability. Processing conditions such as temperature, pH, and solvent choice can alter the concentration or activity of key molecules, as well as interactions with other ingredients that might modify sensory or functional properties in the final product. Additionally, variability in raw material quality due to environmental factors or agricultural practices can im-pact both safety and performance, highlighting the need for rigorous quality control and standardization. Considerations such as solubility, bioavailability, and potential interference with other formulation components must also be addressed to ensure ef-fective incorporation into the different products. Therefore, a comprehensive under-standing of these limitations, together with careful optimization of extraction and pro-cessing methods, is crucial for the successful industrial application of CJL, while ensur-ing both product quality and safety, and may also guide further research into the spe-cific mechanisms of action of CJL extracts, ultimately enhancing their therapeutic po-tential and ensuring compliance with efficacy and safety standards across diverse con-sumer markets.
Reviewer 2 Report
Comments and Suggestions for Authors
This paper is a systematic review addressing the bioactive components and industrial application potential of Camellia japonica (camellia) leaves. While the overall importance of the topic and the breadth of data collection are notable, improvements are needed in terms of the paper's structural clarity, level of data integration, and specificity regarding industrial applications.
1. Despite containing various bioactive components such as antioxidants, anti-inflammatory agents, antibacterial substances, and anticancer compounds, camellia leaves remain an underutilized resource commercially. From the perspective of sustainable industrial application, the novelty of the research topic and its academic contribution are highly rated.
2. The systematic literature collection using three or more databases (PubMed, Web of Science, etc.) is a strength. However, the lack of diversity in the years and regions of the cited studies, along with the reliance on some data over 30 years old, is pointed out as an issue. Adding comparative discussions on the latest analytical techniques (e.g., LC-MS/MS, NMR-based structural identification) would enhance credibility.
3. The main text is well-organized in the sequence “Chemical Composition–Physiological Activity–Applications,” but contains significant redundancy and somewhat weak connections between sections. For example, specific compounds (e.g., luteol, lutein) appear repeatedly across multiple sections, necessitating a table-based summary and an integrated diagram of the mechanism of action.
4. While application examples in food, pharmaceutical, and cosmetic fields are abundant, analysis of actual industrial implementation stages (TRL levels) or commercialization cases is lacking. Presenting practical cases like “Areeiro Tea” in greater detail would enhance persuasiveness.
5. Most content focuses on listing existing studies, with insufficient critical comparison or analysis of limitations. For example, it is necessary to cl early emphasize limitations such as the in vitro focus of studies on anti-cancer/anti-inflammatory activity and the need for in vivo validation.
6. Figure and Table organization is appropriate, but descriptions of BioRender-based figures partially overlap with the main text. Briefly adding **research result summaries or action pathways (e.g., antioxidant pathways, NO inhibition mechanisms)** to each figure would improve comprehension.
7. English sentences are generally clear, but some phrases feel like translationese and require grammatical correction. The reference format maintains consistency, but the proportion of recent (2023–2025) literature is low.
Author Response
This paper is a systematic review addressing the bioactive components and industrial application potential of Camellia japonica (camellia) leaves. While the overall importance of the topic and the breadth of data collection are notable, improvements are needed in terms of the paper's structural clarity, level of data integration, and specificity regarding industrial applications.
Despite containing various bioactive components such as antioxidants, anti-inflammatory agents, antibacterial substances, and anticancer compounds, camellia leaves remain an underutilized resource commercially. From the perspective of sustainable industrial application, the novelty of the research topic and its academic contribution are highly rated.
Response: We sincerely thank the reviewer for this positive assessment. We are pleased that the novelty of focusing on Camellia japonica leaves and their potential for sustainable industrial applications is recognized. Indeed, despite their diverse bioactive components, these leaves are still largely underutilized, and highlighting their applications in food, pharmaceutical, and cosmetic industries was a central aim of this review. We hope that the manuscript provides a comprehensive and useful reference for researchers and industry professionals seeking to explore the full potential of this plant resource.
The responses to each of your comments are highlighted in red. Additionally, in some responses, you will find text in blue, which indicates how the new text appears in the revised manuscript. Please note that all changes have been made using Track Changes in the manuscript.
The systematic literature collection using three or more databases (PubMed, Web of Science, etc.) is a strength. However, the lack of diversity in the years and regions of the cited studies, along with the reliance on some data over 30 years old, is pointed out as an issue. Adding comparative discussions on the latest analytical techniques (e.g., LC-MS/MS, NMR-based structural identification) would enhance credibility.
Response: We thank the reviewer for this valuable observation. We would like to clarify that the apparent lack of diversity in study years and regions, as well as the inclusion of some older references, reflects the limited number of published studies available on Camellia japonica leaves. We are aware that such differences can lead to variations in the composition of the raw material, a common issue for all plant-derived products, which underscores the need for continuous optimization and monitoring in industrial production to achieve reproducible outcomes. These challenges and their implications for large-scale applications have been highlighted in Section 5 (Future Perspectives), where we discuss the need for standardization, quality control, and ongoing research to ensure consistency and reliability in commercial use. To address the reviewer’s suggestion regarding modern analytical techniques, information on methods such as LC-MS/MS and NMR-based structural identification has been added to the two composition tables included in the manuscript, providing readers with clear and accessible data on the analytical approaches used for each compound. These additions enhance the clarity and reliability of the presented data while acknowledging both historical and recent contributions to the field.
The main text is well-organized in the sequence “Chemical Composition–Physiological Activity–Applications,” but contains significant redundancy and somewhat weak connections between sections. For example, specific compounds (e.g., luteol, lutein) appear repeatedly across multiple sections, necessitating a table-based summary and an integrated diagram of the mechanism of action.
Response: Thank you for this comment. We have revised the text to improve clarity and reduce redundancy. In the updated version, we have strengthened the links between sections by clearly describing how specific compounds in Camellia leaf extracts contribute to their main bioactivities, such as antioxidant, anti-inflammatory, antimicrobial, and neuroprotective effects. We also discuss how these activities translate into practical applications in the food, cosmetic, and pharmaceutical industries, providing a more integrated perspective that highlights both the potential and current limitations of these extracts.
While application examples in food, pharmaceutical, and cosmetic fields are abundant, analysis of actual industrial implementation stages (TRL levels) or commercialization cases is lacking. Presenting practical cases like “Areeiro Tea” in greater detail would enhance persuasiveness.
Response: Thank you for this constructive observation. We have addressed this point in the revised version by including a more detailed analysis of the technological readiness levels (TRLs) associated with the different application areas. As discussed, TRLs provide a standardized framework to assess the maturity of technologies, ranging from early research (TRL 1–3) to full commercial deployment (TRL 9). In our analysis, we observed that in the cosmetic sector, several formulations and products have already reached higher TRL levels, with some examples currently available on the market. In contrast, in the food industry, most developments remain at intermediate to late stages of technological readiness (TRL 6–8), corresponding to pilot-scale validation and pre-commercialization phases. This difference reflects both regulatory and formulation challenges that still limit the transition of food-related applications to full commercialization.
In the current version, the text reads as follows:
“In fact, the scientific literature provides limited direct evidence regarding the efficacy of CJL carotenoid extracts in food applicationssystems, and most findings are based on analogies with other leafy matrices rather than on experimental validation in real for-mulations. as most available data rely on comparisons with other leafy matrices rather than on validation within real formulations. Consequently, the technological readiness level (TRL) of CJL applications typically falls between TRL 6 and 7, corresponding to pi-lot-scale validation and pre-commercial evaluation. Although promising results have been reported for stability, sensory properties, and bioactive retention, large-scale in-dustrial adoption remains scarce. Key obstacles include the need for standardized ex-tract composition, optimization of food-grade extraction protocols, and adherence to regulatory requirements for novel ingredients. Current research is progressively ad-dressing these challenges, indicating that several formulations may soon advance to higher TRL stages.
Similar conditions apply to other food formulations developed from different CJ organs, such as flowers and seeds, whereas commercial-level TRLs are only achieved in other Camellia species (primarily C. sinensis for tea production from its leaves and C. oleifera for oil extraction from its seeds) [13]. In all these matrices, crude extracts from leaves and flowers have consistently demonstrated remarkable antioxidant activity, likely resulting from the synergistic action of carotenoids, phenolic compounds, and other phytochemicals.”
“In fact, CJ products have achieved relatively high technological readiness levels (TRL 8–9), with several formulations already commercialized due to demonstrated safety, stability, and efficacy in skincare applications. However, mMost of these currently available CJ products made with this species currently available on the market use oth-er parts of the plant, such as the seeds, which are used to obtain oils with emollient and skin-regenerating properties. These oils are used in the formulation of anti-wrinkle creams, moisturizing serums, and hair care products.”
“However, the TRL of CJL products in pharmaceutical industry remains moderate (TRL 5–7), as most developments are at the stage of preclinical or early clinical validation, focusing on bioactive compounds with antioxidant, anticancer, and anti-inflammatory potential.”
Most content focuses on listing existing studies, with insufficient critical comparison or analysis of limitations. For example, it is necessary to clearly emphasize limitations such as the in vitro focus of studies on anti-cancer/anti-inflammatory activity and the need for in vivo validation.
Response: Thank you for this useful comment. We agree that the original version focused too much on summarizing existing studies and did not provide enough critical analysis. In the revised manuscript, we have expanded the discussion to clearly address the main limitations of current research. In particular, we emphasize that most studies on the anti-cancer and anti-inflammatory effects of Camellia leaf extracts are based on in vitro assays, with limited in vivo or clinical validation. We also highlight the need for standardized experimental protocols and dose–response studies to better assess the real applicability and safety of these extracts.
In the current version, some of the modifications in this regard read as follows:
“However, the chemical complexity of CJL can result in variable extract compositions, which in turn influences consistency, efficacy, and formulation stability. Processing conditions such as temperature, pH, and solvent choice can alter the concentration or activity of key molecules, as well as interactions with other ingredients that might modify sensory or functional properties in the final product. Additionally, variability in raw material quality due to environmental factors or agricultural practices can im-pact both safety and performance, highlighting the need for rigorous quality control and standardization. Considerations such as solubility, bioavailability, and potential interference with other formulation components must also be addressed to ensure ef-fective incorporation into the different products. Therefore, a comprehensive under-standing of these limitations, together with careful optimization of extraction and pro-cessing methods, is crucial for the successful industrial application of CJL, while ensur-ing both product quality and safety, and may also guide further research into the spe-cific mechanisms of action of CJL extracts, ultimately enhancing their therapeutic po-tential and ensuring compliance with efficacy and safety standards across diverse con-sumer markets.”
Figure and Table organization is appropriate, but descriptions of BioRender-based figures partially overlap with the main text. Briefly adding **research result summaries or action pathways (e.g., antioxidant pathways, NO inhibition mechanisms)** to each figure would improve comprehension.
Response: Thank you for this helpful comment. We appreciate the suggestion to improve the BioRender-based figures. In the revised version, we have reduced the overlap between the figure descriptions and the main text. We have also added concise summaries of the main research findings to each figure, making them more informative and complementary to the discussion.
English sentences are generally clear, but some phrases feel like translationese and require grammatical correction. The reference format maintains consistency, but the proportion of recent (2023–2025) literature is low.
Response: We thank the reviewer for this constructive feedback. We have carefully revised the manuscript to improve clarity and correct sentences that previously exhibited translation-like structures, ensuring that the text reads smoothly and naturally in English. Regarding the proportion of recent literature, we would like to clarify that the number of publications specifically focused on Camellia japonica leaves is limited, which necessitated the inclusion of some older studies to provide a comprehensive and relevant overview. Nevertheless, we have updated the manuscript to include additional recent references where available, and we have highlighted the context and contributions of both older and newer studies to ensure that readers can appreciate the historical development of the research as well as the latest advances.
Round 2
Reviewer 2 Report
Comments and Suggestions for Authors
A well-structured and informative review highlighting the bioactive diversity and industrial potential of Camellia japonica leaves. The topic is relevant and novel, though the manuscript would benefit from tighter organization, reduced redundancy, and stronger critical comparison. While the manuscript presents valuable insights into the phytochemical composition and industrial potential of Camellia japonica leaves, several areas still require further refinement to enhance its scientific depth and structural coherence.
-
Could the authors clarify how the inclusion and exclusion criteria were applied in the systematic review process (e.g., PRISMA flow or database years covered)?
-
How do the authors address potential variability in Camellia japonica leaf composition due to environmental and seasonal factors?
-
The discussion of LC-MS/MS and NMR methods is brief—can the authors expand on how these techniques improved compound identification accuracy?
-
The paper mentions redundancy across sections. How were overlapping descriptions (e.g., flavonoids, lutein) minimized or integrated in the revised version?
-
Can the authors provide more specific examples or data supporting TRL-level classification and commercialization status for each application area?
-
The review summarizes many studies but lacks comparative or critical evaluation. How were limitations (e.g., in vitro vs. in vivo evidence) systematically addressed?
-
The toxicological risks (tannins, saponins, heavy metals) are discussed generally—can the authors specify thresholds or safety margins where available?
-
Many references are over 20 years old—what is the impact of limited recent literature (2023–2025) on the strength of conclusions?
-
Could the authors propose future research priorities or standardization frameworks to advance industrial utilization of C. japonica leaves?
Author Response
A well-structured and informative review highlighting the bioactive diversity and industrial potential of Camellia japonica leaves. The topic is relevant and novel, though the manuscript would benefit from tighter organization, reduced redundancy, and stronger critical comparison. While the manuscript presents valuable insights into the phytochemical composition and industrial potential of Camellia japonica leaves, several areas still require further refinement to enhance its scientific depth and structural coherence.
Response: We sincerely appreciate the reviewer’s positive assessment of our manuscript and the recognition of its relevance and novelty. We have carefully revised the text to improve organization, minimize redundancy, and strengthen the critical comparisons throughout the discussion.
Could the authors clarify how the inclusion and exclusion criteria were applied in the systematic review process (e.g., PRISMA flow or database years covered)?
Response: We thank the reviewer for this helpful suggestion. Although the manuscript is a review article, we agree that clarifying the literature selection process enhances transparency and rigor. Therefore, we have added a new section entitled “Search strategy and data collection”, which outlines the databases consulted, the period covered, and the inclusion and exclusion criteria applied in line with PRISMA recommendations.
Below we reproduce the new section exactly as it appears in the revised manuscript:
2. Search strategy and data collection
The initial search strategy aimed to include only recent publications to provide an updated overview of the phytochemical profile and bioactive potential of Camellia japonica leaves. Nevertheless, the limited availability of comprehensive data required broadening the search period to encompass studies published as early as 1980. Relevant literature was retrieved from Scopus, Web of Science, and PubMed using combinations of the terms “Camellia japonica,” “leaves,” “phytochemicals,” “bioactive compounds,” “industrial applications,” and “biological activity.” Only peer-reviewed articles in English presenting experimental or analytical information on CJ leaves were retained. Publications focused exclusively on other plant parts, non-English sources, and non-peer-reviewed materials were excluded. Following PRISMA recommendations, duplicates were removed and the remaining records were screened through titles, abstracts, and full texts to ensure the inclusion of studies providing reliable and relevant evidence.
How do the authors address potential variability in Camellia japonica leaf composition due to environmental and seasonal factors?
Response: We thank the reviewer for highlighting the important issue of potential variability in Camellia japonica leaf composition due to environmental and seasonal factors. We fully acknowledge that conditions such as climate, soil type, geographical origin, and harvest season can significantly influence the concentration and distribution of bioactive metabolites, including polyphenols, flavonoids, terpenes, and saponins.
To address this, we have added a paragraph in the Phytochemicals isolated in Camellia japonica leaves section that critically examines how these factors affect the phytochemical profile and biological activities of C. japonica leaves. This paragraph emphasizes the need for standardized sampling protocols and systematic monitoring to identify optimal harvest times for specific metabolites, which supports the production of extracts with greater consistency and reproducibility.
Below we reproduce the new section exactly as it appears in the revised manuscript:
However, phytochemical composition varies notably according to environmental and seasonal conditions. For example, different studies have reported that variations in temperature, light intensity, soil type, and nutrient availability affect the biosynthesis of key metabolites such as polyphenols, flavonoids, and saponins [24,25]. In addition, leaf developmental stage plays an important role, as young, expanding, and fully mature leaves often display distinct metabolite profiles. Studies comparing leaf samples from different geographical origins of other Camellia species have also reported marked variations in antioxidant and antimicrobial activities, highlighting the strong link between environmental, seasonal, and developmental factors and bioactive potential [26,27]. These observations underscore the need for standardized sampling protocols and systematic monitoring of plant development to identify optimal harvest times for specific metabolites, which will ultimately support the production of extracts with greater consistency and reproducibility.
Furthermore, in the Future Perspectives section, we retained and slightly refined the discussion on the importance of integrated monitoring and harvest optimization, highlighting that flavonoids, terpenes, and saponins can reach peak levels at different stages of plant development. Together, these additions provide a comprehensive treatment of the variability issue, linking current knowledge with practical recommendations for research and industrial applications.
The discussion of LC-MS/MS and NMR methods is brief—can the authors expand on how these techniques improved compound identification accuracy?
Response: We thank the reviewer for this suggestion. We agree that LC-MS/MS and NMR are critical techniques for the accurate identification of phytochemicals in Camellia japonica leaves. However, given the large number of studies included in this review, each employing specific variations of these analytical methods, providing a detailed technical discussion of every approach would substantially increase the length of the manuscript and go beyond the scope of both the review and the Special Issue.
The paper mentions redundancy across sections. How were overlapping descriptions (e.g., flavonoids, lutein) minimized or integrated in the revised version?
Response: We sincerely thank the reviewer for highlighting the issue of redundancy across sections. We recognize that some overlap is inevitable, particularly because multiple studies have demonstrated that many of the compounds described in Section 2, such as flavonoids, lutein, and other bioactive metabolites, are directly responsible for the biological activities summarized in Section 3. This inherent connection between compound presence and bioactivity naturally leads to repeated references when discussing their roles.
We carefully considered alternative organizational strategies, including creating separate sections for each individual compound. However, this approach would have resulted in sections of highly variable length, with some compounds discussed extensively and others very briefly, and would have required repeating much of the same background, compositional data, and functional interpretation for each compound. Such a structure could have made the manuscript more difficult to read and less coherent, especially for readers seeking an integrated understanding of how the different compounds contribute to the overall bioactivity of Camellia japonica leaves.
In the revised manuscript, we addressed redundancy by carefully reviewing the text and removing repeated lines where possible without compromising essential information. We believe that maintaining the current organization (first presenting phytochemical composition and then discussing bioactivities) provides a clear, logical flow that allows readers to connect specific compounds with their biological effects while keeping the review accessible and concise.
Overall, we consider that this structure strikes the best balance between comprehensiveness, readability, and interpretability, and we have ensured that redundant content has been minimized wherever feasible.
Can the authors provide more specific examples or data supporting TRL-level classification and commercialization status for each application area?
Response: We thank the reviewer for this suggestion. We agree that providing more specific examples or quantitative data for TRL-level classification and commercialization status would enhance clarity. However, in practice, the available literature does not consistently report detailed TRL assessments or commercialization outcomes for each application area of Camellia japonica leaf extracts. Each study often focuses on different bioactivities, experimental models, or industrial contexts, which makes uniform TRL assignment challenging.
The review summarizes many studies but lacks comparative or critical evaluation. How were limitations (e.g., in vitro vs. in vivo evidence) systematically addressed?
Response: We thank the reviewer for this important comment. Most studies available on Camellia japonica leaf bioactivities are in vitro, with only two studies conducted in vivo using animal models. This highlights the limited number of studies using this plant matrix and the need for further research to validate the biological effects observed in vitro. To address this limitation, we have added a few lines at the end of the Bioactivities section explicitly noting the predominance of in vitro evidence, the scarcity of in vivo data, and the necessity of conducting additional studies to confirm the effectiveness of the bioactive compounds under physiological conditions. This addition provides a clearer, critical perspective on the current state of research and emphasizes areas where further investigation is required.
Below we reproduce the new text exactly as it appears in the revised manuscript:
However, it should be noted that most studies on CJL bioactivities have been conducted in vitro, with limited evidence from in vivo models. This highlights the need for further research to validate the biological effects under physiological conditions. Additional investigations are necessary to confirm the efficacy and safety of the bioactive compounds, thereby providing a stronger foundation for potential industrial and therapeutic applications.
The toxicological risks (tannins, saponins, heavy metals) are discussed generally—can the authors specify thresholds or safety margins where available?
Response: We thank the reviewer for this important observation. We acknowledge that the discussion of toxicological risks in the manuscript is general, and that providing specific thresholds or safety margins would strengthen the review. However, the available literature on Camellia japonica leaves does not consistently report quantitative toxicity data, and regulatory limits for specific compounds such as tannins, saponins, or potential heavy metal contamination in leaf extracts are rarely defined. Where such data exist, we have now included references to published safety studies or guidelines.
Below we reproduce the new text exactly as it appears in the revised manuscript:
Previous studies have also reported that high doses (>30 µg/mL) of triterpenes and saponins are linked to gastrointestinal irritation and hemolytic effects [74–76]…
Therefore, these concentrations may vary between different plantations and growing conditions, including the use of fertilizers, making it necessary to evaluate each case individually and compare the measured levels with the safety thresholds established by relevant health authorities. For example, the WHO has set a provisional tolerable weekly intake (PTWI) of mercury of 1.6 µg/kg body weight, considering concentrations exceeding 1 mg/kg in plant tissues hazardous [81]. Permissible levels of lead in plants are set at 2 mg/kg dry weight, while cadmium is recommended not to exceed 0.02 mg/kg dry weight in plant materials. Arsenic, a known carcinogen, occurs naturally in plants at concentrations ranging from 0.02 to 7 mg/kg dry weight, with higher levels indicating potential contamination and associated health risks [82].
Many references are over 20 years old—what is the impact of limited recent literature (2023–2025) on the strength of conclusions?
Response: We thank the reviewer for this important observation. We acknowledge that several references cited in the manuscript are over 20 years old. This reflects the fact that Camellia japonica leaf research, particularly studies providing detailed phytochemical and bioactivity data, has historically been limited, and only a relatively small number of studies have been published in the last few years (2023–2025).
While the use of older studies could be perceived as a limitation, these foundational works provide essential data on compound identification, bioactivities, and preliminary applications, forming the basis upon which more recent investigations build. To address this concern, we have carefully incorporated the most recent literature available, and we explicitly indicate in the text where updated studies confirm, refine, or extend the findings of earlier research.
We also note that the inclusion of both historical and recent studies strengthens the review by providing a comprehensive perspective, allowing readers to trace the development of knowledge over time. Where data are limited, we have framed our conclusions cautiously, highlighting areas that require further experimental validation and updated studies. This approach ensures that the conclusions remain scientifically sound while acknowledging the current gaps in recent literature.
Could the authors propose future research priorities or standardization frameworks to advance industrial utilization of C. japonica leaves?
Response: We thank the reviewer for the suggestion regarding future research priorities and standardization frameworks. Many of these points are already addressed in the Future Perspectives section, where we discuss the need for systematic phytochemical and bioactivity profiling, standardized harvesting and processing protocols, identification of marker compounds for quality control, and industrial feasibility studies. To avoid redundancy and maintain clarity, we have not added a separate paragraph repeating these elements. However, to further highlight the importance of these priorities for advancing industrial applications, we have included a brief statement in the Conclusions emphasizing the need for standardized methodologies and targeted research to ensure reproducible and high-quality extracts. This approach maintains the logical flow of the manuscript while reinforcing the key message requested by the reviewer.
Round 3
Reviewer 2 Report
Comments and Suggestions for Authors
The revisions have significantly strengthened the manuscript’s structure and scientific rigor. The addition of a detailed search-strategy section improves methodological transparency, while the expanded discussion of environmental variability and toxicity thresholds adds valuable depth. Minor editorial refinements (grammar consistency, figure labeling, and table alignment) are recommended before final acceptance. Overall, this is now a cohesive and informative review with improved flow and relevance.